# Modeling the impact of heterogeneous reactions of chlorine on summertime nitrate formation in Beijing, China

Xionghui Qiu[1,2], Qi Ying[3*], Shuxiao Wang [1,2*], Lei Duan[1,2], Jian Zhao[4], Jia Xing[1,2], Dian Ding[1,2], Yele Sun[4], Baoxian Liu[5], Aijun Shi[6], Xiao Yan[6], Qingcheng Xu[1,2], Jiming Hao[1,2]

[1] State Key Joint Laboratory of Environmental Simulation and Pollution Control, School of Environment, Tsinghua University. Beijing 100084, China.

[2] State Environmental Protection Key Laboratory of Sources and Control of Air Pollution Complex, Beijing 100084, China

[3] Zachry Department of Civil Engineering, Texas A&M University, College Station, Texas 77843-3138, United States

[4] State Key Laboratory of Atmospheric Boundary Layer Physics and Atmospheric Chemistry, Institute of Atmospheric Physics, Chinese Academy of Sciences, Beijing 100029, China

[5] Beijing Environmental Monitoring Center, Beijing 100048, China

[6] Beijing Municipal Research Institute of Environmental Protection, Beijing 100037, China

*Corresponding author: shxwang@tsinghua.edu.cn & qying@civil.tamu.edu

Abstract:

A comprehensive chlorine heterogeneous chemistry is incorporated into the Community Multiscale Air Quality (CMAQ) model to evaluate the impact of chlorine-related heterogeneous reaction on diurnal and nocturnal nitrate formation and quantify the nitrate formation from gas-to-particle partitioning of $HNO_3$ and from different heterogeneous pathways. The results show that these heterogeneous reactions increase the atmospheric $Cl_2$ and $ClNO_2$ level (~100%), which further affect the nitrate formation. Sensitivity analyses of uptake coefficients show that the empirical uptake coefficient for the $O_3$ heterogeneous reaction with chlorinated particles may lead to the large uncertainties in the predicted $Cl_2$ and nitrate concentrations. The $N_2O_5$ uptake

coefficient with particulate $Cl^-$ concentration dependence performs better to capture the
concentration of $ClNO_2$ and nocturnal nitrate concentration. The reaction of OH and
$NO_2$ in daytime increases the nitrate by ~15% when the heterogeneous chlorine
chemistry is incorporated, resulting in more nitrate formation from $HNO_3$ gas-to-
particle partitioning. By contrast, the contribution of the heterogeneous reaction of
$N_2O_5$ to nitrate concentrations decreases by about 27% in the nighttime when its
reactions with chlorinated particles are considered. However, the generated gas-phase
$ClNO_2$ from the heterogeneous reaction of $N_2O_5$ and chlorine-containing particles
further reacts with the particle surface to increase the nitrate by 6%. In general, this
study highlights the potential of significant underestimation of daytime and
overestimation of nighttime nitrate concentrations for chemical transport models
without proper chlorine chemistry in the gas and particle phases.

**1.  Introduction**
In recent years, nitrate has become the primary component of $PM_{2.5}$ (particulate matter
with aerodynamic diameter less than 2.5μm) in Beijing with sustained and rapid
reduction of $SO_2$ and primary particulate matter emissions (Ma et al., 2018; Li et al.,
2018; Wen et al., 2018). Observations showed that the relative contributions of
secondary nitrate in $PM_{2.5}$ could reach up to approximately 50% during some severe
haze pollution days (Li et al., 2018). The mechanism of secondary nitrate formation can
be summarized as two major pathways: (1) Gas-to-particle partitioning of $HNO_3$, which
happens mostly in the daytime. The reaction of OH with $NO_2$ produces gaseous $HNO_3$,
which subsequently partition into the particle phase. The existence of $NH_3$ or basic
particles enhances this process by $NH_3$–$NH_4^+$ gas-particle equilibrium (Kleeman et al.,
2005; Seinfeld and Pandis, 2006); (2) Hydrolysis of $N_2O_5$, which is more important at
nighttime. $N_2O_5$ forms from the reactions of $NO_2$, $O_3$ and $NO_3$ and hydrolyzes to
produce particulate nitrate. They can be summarized as reactions R1-R5 (Brown and
Stutz 2012):

$OH + NO_2 \rightarrow HNO_3$                    (R1)

$$HNO_3 + NH_3 \rightarrow NH_4^+ + NO_3^- \qquad (R2)$$
$$NO_2 + O_3 \rightarrow NO_3 + O_2 \qquad (R3)$$
$$NO_3 + NO_2 \leftrightarrow N_2O_5 \qquad (R4)$$
$$N_2O_5 + H_2O(aq) \rightarrow 2H^+ + 2NO_3^- \qquad (R5)$$
In addition to reactions R1 and R5, gas phase reactions of $NO_3$ with $HO_2$ and VOCs,
$N_2O_5$ with water vapor (Tuazon et al., 1983) and the heterogeneous reaction of $NO_2$
with water-containing particle (Goodman et al., 1999) produce $HNO_3$ or nitrate as well.
Theses reactions are listed in Table 2 as reactions R8, R9 and R10.

However, chemistry transport models (CTMs) incorporated with these mechanisms

still can't accurately capture the spatiotemporal distributions of nitrate in some studies
in polluted northern China. For example, Chang et al. (2018) showed that the simulated
nitrate concentrations derived from the default CMAQ (version 5.0.2) were
significantly higher than the observations in summer at two sites adjacent to Beijing.
Fu et al. (2016) also found that default CMAQ (version 5.0.1) overestimated the
simulated nitrate concentrations in the Beijing-Tianjin-Hebei region.

In recent fields studies, it was found that high particulate chlorine emissions might

have a significant impact on the oxidation capacity of the urban atmosphere and thus
could affect nitrate concentrations. According to the field measurements in June 2017
in Beijing (Zhou et al., 2018), the 2-min averaged concentrations of reactive molecular
chlorine ($Cl_2$) and nitryl chloride ($ClNO_2$) reached up to 1000 pptv and 1200 pptv,
respectively, during some severe air pollution periods in summer. The $Cl_2$
concentrations were significantly higher than those observed in North American coastal
cities affected by onshore flow and the lower atmosphere in the remote Arctic region
(Spicer et al., 1998; Glasow et al., 2010; Liu et al., 2017). During these pollution events,
the corresponding concentrations of $N_2O_5$ (2-min average) and nitrate (5-min average)
rose from 40 pptv and 1 $\mu g\ m^{-3}$ to 700 pptv and 5 $\mu g\ m^{-3}$. To explain the high levels of
$ClNO_2$, some studies suggested that reaction R5 should be revised to account for $ClNO_2$
production from the heterogeneous reaction of $N_2O_5$ on chloride-containing particles
(CPS) (Osthoff et al., 2008; Thornton et al., 2010), as shown in reaction R6:
$$N_2O_5 + (1-\phi)\ H_2O + \phi\ Cl^- \rightarrow \phi\ ClNO_2 + (2-\phi)\ NO_3^- \qquad (R6)$$

where $\phi$ represents the molar yield of $ClNO_2$. By incorporating this reaction into WRF-Chem, Li et al. (2016) found that the improved model performed better to match the observed nitrate concentrations in Hongkong during 15 November and 5 December 2013. However, $ClNO_2$ could affect the formation of nitrate indirectly by increasing the atmospheric OH after a series of chemical reactions, which are briefly summarized into three steps: (1) the photolysis of $ClNO_2$ produces chlorine radicals ($Cl^\bullet$); (2) the reaction of $Cl^\bullet$ with VOCs produces peroxy radical ($HO_2$ and $RO_2$); and (3) the increased $HO_2$ and $RO_2$ prompt the formation of OH through HOx cycle and lead to increased $HNO_3$ production (Young et al., 2014; Jobson et al., 1994). The overall impact of R6 on nitrate remains to be investigated.

Another related but unresolved issue is the sources of the high concentrations $Cl_2$, which could not be explained by the $N_2O_5$ heterogeneous reaction with $Cl^-$ and the subsequent reactions of $ClNO_2$ in the gas phase. It has been reported that the reactions of gaseous $O_3$, OH, $HO_2$, $ClNO_2$, hypochlorous acid (HOCl), chlorine nitrate ($ClONO_2$) with CPS can produce $Cl_2$, which can subsequently photolyze to produce $Cl^\bullet$ (Knipping et al., 2000; George et al., 2010; Pratte et al., 2006; Deiber et al., 2004; Faxon et al., 2015). However, these heterogeneous reactions on CPS are generally missing in most of the current CTMs and it is unclear whether these reactions will be able to explain the observed $Cl_2$ concentrations and the overall impact of these reactions on nitrate.

Previously, biomass burning, coal combustion, and waste incineration were identified as the main sources of gaseous and particulate chlorine compounds in China from International Global Atmospheric Chemistry Program's Global Emissions Inventory Activity (GEIA) based on the year 1990 and a localized study by Fu et al. based on the year 2014 (Keene et al., 1999; Fu et al., 2018). However, recent source apportionment results of $PM_{2.5}$ in Beijing showed that the contribution of coal combustion had extremely decreased from 22.4% in 2014 to 3% in 2017 with the replacement of natural gas (obtained from official website of Beijing Municipal Bureau of Statistics, available at http://edu.bjstats.gov.cn/). Another important source—cooking has received attention as its increasing contribution to $PM_{2.5}$ (accounting for 33% of the

residential sector; obtained from the official source apportionment analysis of $PM_{2.5}$ in
Beijing in 2017; see http://www.bjepb.gov.cn/bjhr-b/index/index.html). Moreover, the
high content of particulate sodium chloride was measured from the source
characterization studies of $PM_{2.5}$ released from the cooking activities (Zhang et al.,
2016). Thus, it is necessary to compile an updated emission inventory for Beijing to
include the emissions from cooking and other sources (coal burning, solid waste
incineration, biomass burning, etc.) in order to explore the emissions of the chlorine
species on atmospheric nitrate formation.
In this study, a CMAQ model with an improved chlorine heterogeneous chemistry
is applied to simulate summer nitrate concentration in Beijing. Sensitivity simulations
are conducted to evaluate the contributions of $HNO_3$ gas-to-particle partitioning and
heterogonous production to aerosol nitrate. The results of this work can improve our
understandings on nitrate formation and provide useful information on nitrate pollution
control strategies in Beijing.

**2. Emissions, chemical reactions and model description**
2.1 Emissions
Generally, the conventional emission inventories of air pollutants in China only include
the common chemical species, such as $SO_2$, $NO_X$, VOCs, $PM_{2.5}$, $PM_{10}$, $NH_3$, BC, and
OC (Wang et al., 2014). Chloride compound emissions were not included. However,
the emissions of chlorine species are vital for studying the chlorine chemical
mechanism. Recently, the inorganic hydrogen chloride (HCl) and fine particulate
chloride (PCl) emission inventories for the sectors of coal combustion, biomass burning,
and waste incineration were developed for the year of 2014 (Qiu et al., 2016, Fu et al.,
2018, Liu et al., 2018). However, the gaseous chlorine emission was not estimated in
these studies. In addition, these studies did not account for the rapid decrease of coal
consumption in recent years in Beijing, from 2000 Mt in 2014 to 490 Mt in 2017. More
importantly, the cooking source, as one of the major contributors to particulate chlorine
in Beijing, is not included in current chlorine emission inventories. Thus, a new
emission inventory of reactive chlorine species, which includes HCl, $Cl_2$ and PCl, were
developed in this study for the year of 2017.

The emission factor method (equation (1)) is applied to calculate the emissions of

these reactive chlorine species from coal combustion, biomass burning, municipal solid
waste incineration and industrial processes:
$$E_{i,j} = A_i \times EF_{i,j} \tag{1}$$
where $E_{i,j}$ represents the emission factor of pollutant $j$ in sector $i$; $A$ represents the
activity data; $EF$ represents the emission factor. $EF$ for PCl is estimated by $EF_{i,PCl} =$
$EF_{i,PM2.5} \times f_{Cl,i}$, where $f_{Cl,i}$ represents the mass fraction of PCl in primary $PM_{2.5}$.
Activity data are obtained from the Beijing Municipal Bureau of Statistics (available at
http://tjj.beijing.gov.cn/). The $Cl_2$ emission factor for coal combustion is calculated
based on the content of Cl in coal, which had been measured by Deng et al (2017). The
$PM_{2.5}$ emission factors and mass fractions of PCl in $PM_{2.5}$ to calculate the emissions of
Cl had been described in detail by Fu et al. (2018). PCl in $PM_{2.5}$ for coal combustion
and biomass burning are taken as 1% and 9.0%, respectively, based on local
measurements in Beijing.

Emissions of PCl from cooking, including contributions from commercial and

household cooking, are estimated using equation (2):
$$E_{PCl} = \left[N_f \times V_f \times H_f \times EF_{f,PCl} + V_c \times H_c \times N_c \times n \times EF_{c,PCl} \times (1-\eta)\right] \times 365 \tag{2}$$
where $N_f$ is the number of households, $V_f$ is the volume of exhaust gas from a household
stove (2000 $m^3$ $h^{-1}$); $H_f$ is the cooking time for a family (0.5 h day$^{-1}$); $EF_{f,PCl}$ and
$EF_{c,PCl}$ are the emission factors (kg m$^{-3}$) of PCl for household and commercial cooking,
respectively; $Hc$ is the cooking time in a commercial cooking facility (6 h day$^{-1}$); $Nc$ is
the number of restaurants, schools and government departments. $Vc$ is the volume of
exhaust gas from a commercial cooking stove (8000 $m^3$ $h^{-1}$); $n$ is the number of stoves
for each unit, which equals to 6 for a restaurant and is calculated as one stove per 150
students for each school. $\eta$ is the removal efficiency of fume scrubbers (30%). $EF_{c,PCl}$
is the emission factor (kg m$^{-3}$) of PCl in commercial cooking. These constants are all
based on Wu et al. (2018). The PCl fraction in $PM_{2.5}$ from cooking is take as 10%, based
on local measurements. HCl and $Cl_2$ emissions from cooking are not considered in this
study.
The sectoral emissions of HCl, $Cl_2$ and PCl are summarized in Table 1. The
estimated HCl, $Cl_2$ and PCl emissions in Beijing are 1.89 Gg, 0.07Gg and 0.63Gg
respectively. The Cl emissions estimated for 2014 by Fu et al. (2018) were used for
other areas. This simplification is a good approximation because replacing coal with
natural gas only occurred in Beijing, and reduction of coal consumption in surrounding
regions was generally less than 15%. In addition, strict control measures for biomass
burning, cooking and municipal solid waste incineration have not been implemented in
most regions yet. Emissions of conventional species for this study period are developed
in a separate study that is currently under review and are summarized in Table S1.

2.2 Chlorine-related heterogeneous reactions
The heterogeneous reactions in original CMAQ (version 5.0.1) are not related to
chlorine species. In this study, the original heterogeneous reactions of $N_2O_5$ and $NO_2$
(R5 and R10 in Table 2) are replaced with a revised version which includes production
of $ClNO_2$ from CPS (R6 and R11 in Table 2). In reaction R6, the molar yield of $ClNO_2$
($\phi_{ClNO_2}$) is represented as equation (3) (Bertram and Thornton, 2009):

$$\phi_{ClNO_2} = \left(1 + \frac{[H_2O]}{483 \times [Cl^-]}\right)^{-1} \qquad (3)$$

where $[H_2O]$ and $[Cl^-]$ are the molarities of liquid water and chloride (mol m$^{-3}$).
In addition, laboratory observations confirmed that the heterogeneous uptake of
some oxidants (such as $O_3$ and OH) and reactive chlorine species (such as $ClNO_2$, HOCl,
and $ClONO_2$) could also occur on CPS to produce $Cl_2$. These reactions are implemented
in the model and summarized in Table 2 as R13-R18. Note that the products from the
heterogeneous uptake of $ClNO_2$ on CPS vary with particle acidity (Riedel et al., 2012;
Rossi, 2003). It generates $Cl_2$ under the condition of pH lower than 2 but produces
nitrate and chloride under higher pH conditions. The reaction rates of the heterogeneous
reactions are parameterized as first-order reactions, with the rate of change of gas phase
species concentrations determined by equations (4) (Ying et al., 2015):

$$\frac{dC}{dt} = -\frac{1}{4}(v\gamma A)C = -k^{\mathrm{I}}C \tag{4}$$

where $C$ represents the concentration of species; $v$ represents the thermal velocity of the gas molecules (m s$^{-1}$); $A$ is the CMAQ-predicted wet aerosol surface area concentration (m$^2$ m$^{-3}$); $\gamma$ represents the uptake coefficient. For all gas phases species (except ClNO$_2$) involved in the heterogeneous reactions (R6 and R11-R18), a simple analytical solution can be used to update their concentrations from time $t_0$ to $t_0+\Delta t$: $[C]_{t0+\Delta t}=[C]_{t0}\exp(-k^{\mathrm{I}}\Delta t)$, where $\Delta t$ is the operator-splitting time step for heterogeneous reactions.

The rate of change of ClNO$_2$ includes both removal and production terms, as shown in equation (5):

$$\frac{d[\mathrm{ClNO_2}]}{dt} = -k_{\mathrm{i}}^{\mathrm{I}}[\mathrm{ClNO_2}] + k_6^{\mathrm{I}}\phi_{\mathrm{ClNO2}}[\mathrm{N_2O_5}] \tag{5}$$
$$= -k_{\mathrm{i}}^{\mathrm{I}}[\mathrm{ClNO_2}] + k_6^{\mathrm{I}}\phi_{\mathrm{ClNO2}}[\mathrm{N_2O_5}]_{t0}\exp(-k_6^{\mathrm{I}}t)$$

Assuming $\phi_{\mathrm{ClNO2}}$ is a constant, an analytical solution for equation (5) can be obtained, as shown in equation (6):

$$[\mathrm{ClNO_2}]_{t0+\Delta t} = [\mathrm{ClNO_2}]_{t0}\exp(-k_{\mathrm{i}}^{\mathrm{I}}\Delta t)$$
$$+ \frac{k_6^{\mathrm{I}}\phi_{\mathrm{ClNO2}}[\mathrm{N_2O_5}]_{t0}}{k_{\mathrm{i}}^{\mathrm{I}} - k_6^{\mathrm{I}}}\left[\exp(-k_6^{\mathrm{I}}\Delta t) - \exp(-k_{\mathrm{i}}^{\mathrm{I}}\Delta t)\right] \tag{6}$$

where $k_{\mathrm{i}}^{\mathrm{I}}$ represents the pseudo first-order rate coefficient of either reaction R17 or R18, depending on pH.

The uptake coefficients $\gamma$ of gaseous species are obtained from published laboratorial studies. In the original CMAQ, the uptake coefficient of N$_2$O$_5$ is determined as a function of the concentrations of (NH$_4$)$_2$SO$_4$, NH$_4$HSO$_4$ and NH$_4$NO$_3$ (Davis et al., 2008). In this study, the PCl and NO$_3^-$ dependent parameterization by Bertram and Thornton (2009) (equation (7)) is used:

$$\gamma_{N_2O_5} = \begin{cases} 0.02, & for\ frozen\ aerosols \\ 3.2 \times 10^{-8}K_f\left[1 - \left(1 + \frac{6 \times 10^{-2}[\mathrm{H_2O}]}{[\mathrm{NO_3^-}]} + \frac{29[\mathrm{Cl^-}]}{[\mathrm{NO_3^-}]}\right)^{-1}\right] \end{cases} \tag{7}$$

In the above equation, $K_f$ is parameterized function based on molarity of water: $K_f = 1.15 \times 10^6(1 - e^{-0.13[\mathrm{H_2O}]})$. NO$_3^-$ and Cl$^-$ concentrations are also in molarity. The uptake coefficient of OH is expressed in equation (8) as a function of the concentration of PCl following the IUPAC (International Union of Pure and Applied Chemistry,

available                        at                        http://iupac.poleether.fr/htdocs/datasheets/pdf/O-
H_halide_solutions_VI.A2.1.pdf).

$$\gamma_{\text{OH}} = \min(0.04 \times \frac{[Cl^-]}{1000 \times M}, 1) \tag{8}$$

where $M$ represents the volume of liquid water in aerosol volume ($m^3$ $m^{-3}$). For frozen
particles, the uptake coefficient is limited to 0.02, as used in the original CMAQ model.
The uptake coefficients of $O_3$, $NO_3$, $NO_2$, HOCl, ClNO$_2$, and ClONO$_2$ are treated
as constants. Among of them, the $\gamma$ values of $NO_3$, $NO_2$, HOCl and ClONO$_2$ are set as
$3 \times 10^{-3}$, $1 \times 10^{-4}$, $1.09 \times 10^{-3}$ and 0.16 based on laboratory measurements (Rudich et al.,
1996; Abbatt et al., 1998; Pratte et al., 2006; Gebel et al., 2001). A preliminary value
of $10^{-3}$ in the daytime and $10^{-5}$ at nighttime is chosen for the $O_3$ uptake coefficient. The
daytime $\gamma_{O_3}$ is based on the analysis of $Cl_2$ production rate in a hypothesized
geochemical cycle of reactive inorganic chlorine in the marine boundary layer by Keene
et al. (1990). The lower nighttime value was also recommended by Keene et al. (1990)
who noted that $Cl_2$ production in the marine boundary layer are lower at night. The
uptake coefficient of ClNO$_2$ depends on the particle acidity, with the value of $2.65 \times 10^{-6}$
$^6$ for reaction R17 and $6 \times 10^{-3}$ for reaction R18 (Robert et al., 2008).

2.4 CMAQ model configuration
These heterogeneous reactions of chlorine are incorporated into a revised CMAQ based
on the CMAQ version 5.0.1 to simulate the distribution of nitrate concentration in
Beijing from 11 to 15 June 2017. The revised CMAQ model without heterogeneous
reactions of chlorine has been described in detail by Ying et al. (2015) and Hu et al.
(2016, 2017). In summary, the gas phase chemical mechanism in the revised CMAQ
model is based on the SAPRC-11 (Cater et al., 2012) with a comprehensive inorganic
chlorine chemistry. Reactions of Cl radical with several major VOCs, which lead to
production of HCl, are also included. The aerosol module is based on AERO6 with an
updated treatment of $NO_2$ and $SO_2$ heterogeneous reaction and formation of secondary
organic aerosol from isoprene epoxides. Three-level nested domains with the
resolutions of 36km, 12km, and 4km using Lambert Conformal Conic projection
(173×136, 135×228 and 60×66 grid cells) are chosen in this work (the domains see
Figure S1). The two true latitudes are set to 25°N and 40°N and the origin of the domain
is set at 34°N, 110°E. The left-bottom coordinates of the outmost domain are positioned
at x = -3114 km, y = -2448 km. The BASE case (heterogeneous reactions of Cl turned
off) and HET case (all heterogeneous reactions enabled) are compared to evaluate the
impact of heterogeneous chlorine chemistry on nitrate formation.

## 3. Results

3.1 Model performance evaluation
Predicted $O_3$, $NO_2$ and $PM_{2.5}$ concentrations from the BASE case simulation are
evaluated against monitoring data at 12 sites in Beijing (Table S2) in 11 to 15, June
2017. The average NMB/NME values for $O_3$, $NO_2$ and $PM_{2.5}$ across the 12 sites are -
8%/29%, -7%/59% and -8%/53%, respectively. Predicted hourly $Cl_2$, $ClNO_2$ and $N_2O_5$
concentrations were compared with observations measured at the Institute of
Atmospheric Physics (IAP), Chinese Academy of Sciences (39.98°N, 116.37°E) using
a high-resolution time-of-flight chemical ionization mass spectrometer (CIMS) from 11
to 15 June 2017 (for site description, instrument introduction, and analytical method,
please refer to the study by Zhou et al. (2018)). Figure 1 shows that the concentrations
of $Cl_2$ and $ClNO_2$ in BASE case are rather low (close to 0), proving that the gas-phase
chemistry is not the major pathway to produce $Cl_2$ and $ClNO_2$. By contrast, the
simulated $Cl_2$ and $ClNO_2$ concentrations in HET case increase significantly,
correspondingly the NMB and NME changes from -100% to -54% and 100% to 61%
for $Cl_2$, and from -100% to -58% and 100% to 62% for $ClNO_2$, respectively (the particle
surface area concentrations is scaled up by a factor of 5 in daytime and 10 in nighttime
because this parameter is underestimated compared to the measured concentrations
reported by Zhou et al. (2018)). The simulations of $Cl_2$ and $ClNO_2$ are improved as the
additional heterogeneous reactions prompt the production of gas phase molecular
chlorine. Overall, however, the $Cl_2$ and $ClNO_2$ concentrations are still underestimated.
Both BASE and HET simulations generally capture the hourly $N_2O_5$ concentrations as
well as the peak values (Figure 1(c)) with similar overall NMB and NME values.
The uptake coefficient of $O_3$ could be an important factor affecting the predicted
$Cl_2$ concentrations as it is found that the heterogeneous reaction of $O_3$ is the major
source of $Cl_2$ during this period (see discussion in Section 3.2). The influence of
different parametrizations of the uptake coefficient of $N_2O_5$ on $ClNO_2$ and nitrate
concentrations are also discussed in Section 3.2.
Predicted $NO_3^-$ and PCl concentrations are compared with observations measured
at an adjacent monitoring site located at the rooftop of School of Environment building
in Tsinghua University (THU, 40.00°N, 116.34°E, about 5 km from IAP) using an
Online Analyser of Monitoring of Aerosol and Gases (MARGA) from 11 to 15 June
2017. According to Figure 1(d), the simulated nitrate concentration is slightly lower
than the observations most of the time. From the evening hours of 12 June to morning
hours of 13 June, observed and simulated nitrate concentration both increase
significantly. The NMB and NME values of hourly nitrate for the HET case (-5% and
39%, respectively) are slightly lower than those for the BASE case -10% and 46%)
during this high concentration period. The HET case also generally captures the day-
to-day variation of PCl concentration and perform better than the BASE case,
correspondingly the NMB and NME are reduced from -48% and 72% to -37% and 67%.
The substantial underestimation of PCl in the daytime on 15 June is likely caused by
missing local emissions during this period.

3.2 Impact of uptake coefficients of $O_3$ and $N_2O_5$ on chlorine species and nitrate
The uptake coefficients of $O_3$ and $N_2O_5$ may be important factors affecting the accuracy
of simulated nitrate concentrations. Some studies have confirmed that the reaction of
$O_3$ on CPS can indirectly affect the nitrate formation by increasing the atmospheric $Cl_2$
and OH level (Li et al., 2016; Liu et al., 2018). According to Figure 1(a), the improved
model still substantially underestimates the concentration of $Cl_2$, which may be
associated with the underestimation of the uptake coefficients of $O_3$, which are
empirical and have not been confirmed by laboratory studies. The uptake coefficients
were increased by a factor of 10 (0.01 for daytime and $10^{-4}$ for nighttime) to evaluate
the sensitivity of $Cl_2$ production and nitrate formation to this parameter. Figure 2 shows
that the simulated $Cl_2$ and nitrate concentrations in daytime increase significantly
(especially for $Cl_2$) and sometimes can capture the peak value (such as the daytime peak
on 14 June). However, although the NMB and NME of $Cl_2$ and nitrate improve from -
18% and 39% to 1% and 28% when the new uptake coefficients are used, the simulated
$Cl_2$ concentrations are still quite different from the observations (such as during the
daytime in 11 and 12 June, see Figure 2). A non-constant parameterization of the uptake
coefficients of $O_3$ that considers the influence of PCl concentrations, meteorology
conditions, etc., similar to those of OH and $N_2O_5$, might be needed. Further laboratory
studies should be conducted to provide a better estimation of this important parameter.

Several parameterizations for the uptake coefficient of $N_2O_5$ have been developed

for regional and global models and have been evaluated in several previous studies
(Tham et al., 2018, McDuffie et al., 2018a, 2018b). In addition to the parameterization
of Bertram and Thornton (2009) used in the HET case, two additional simulations were
performed to assess the impact of the uptake coefficient of $N_2O_5$ on nitrate formation.
The first simulation uses the original CMAQ parameterization of Davis et al.(2008) and
second simulation uses a constant value of 0.09, which is the upper limit of the $N_2O_5$
uptake coefficient derived by Zhou et al. (2018) based on observations. The results from
the simulations with the parameterization of Bertram and Thornton (2009) generally
agree with the results using those based on Davis et al. (2008) .The application of larger
and fixed $N_2O_5$ uptake coefficient leads to slightly better results, which might reflect
the fact that the $N_2O_5$ concentrations are underestimated. Using the uptake coefficient
of 0.09 can generally increase the concentration of nitrate in some periods, but it also
leads to significant increase of the nitrate level (such as nighttime on 12-13 June and
13-14 June), which is 4-6 times higher than those based on Bertram and Thornton
(2009). Overall, predicted nitrate concentrations are sensitive to changes in the changes
in $\gamma_{N_2O_5}$, with approximately 50% increase in the nitrate when a constant of $\gamma_{N_2O_5}$ of
0.09 is used.

3.3 Spatial distributions of nitrate and chlorine species concentrations
The regional distributions of averaged $Cl_2$, $ClNO_2$, $N_2O_5$ and $NO_3^-$ concentration from
11 to 15 June for the HET case are shown in Figure 3. Compared to the BASE case, the

episode average concentrations of $Cl_2$ and $ClNO_2$ from the HET case increase significantly in the eastern region of Beijing, reaching up to 23 ppt and 71 ppt from near zero (Figure 3a and 3b). High concentrations are not found in the southern region with intensive emissions of chlorine species (Figure S2). The production of $ClNO_2$ requires the presence of chloride, $NO_2$, and $O_3$. In the areas close to the fresh emissions, $O_3$ is generally low (Figure S3), and the production of $NO_3$ (hence $N_2O_5$ and $ClNO_2$) is limited. Therefore, the production rate of $ClNO_2$ is generally low in areas affected by fresh emissions. Since the contribution of direct emissions to $Cl_2$ is low and it is predominantly produced secondarily in the atmosphere, high levels of $Cl_2$ are also found away from the fresh emissions.

The spatial distribution of $N_2O_5$ concentrations differs from that of other species (Figure 3c). While the concentrations of most of the species are higher in the southern region, the $N_2O_5$ concentrations are lower in some parts of this region. This is because the $O_3$ concentration in the core urban areas is low due to high $NO_x$ emissions. The $N_2O_5$ concentrations from the HET case are approximately 16% lower on average (Figure 3d) because the Bertram and Thornton (2009) parameterization used in the HET case generally gives higher uptake coefficients than the parameterization of Davis et al. (2008) used in the BASE case (Table 3).

Although the higher uptake coefficients of $N_2O_5$ in the HET case facilitate faster conversion of $N_2O_5$ to nitrate, the nitrate concentrations do not always increase. During daytime hours, nitrate concentrations in the HET case increase due to higher OH (Figure 3e and Figure 3f, increased OH see Figure S4). At nighttime, in contrast, the nitrate concentration decreases significantly in some regions by about 22%, mainly due to lower molar yield of nitrate from the $N_2O_5$ heterogeneous reaction in the HET case (Figure 3g and Figure 3h). Although $ClNO_2$ produced in the $N_2O_5$ reaction also produces nitrate through a heterogenous reaction when the particle pH is above 2, which is true for most regions (see Figure S5), the uptake coefficient of $ClNO_2$ is significantly lower than that of $N_2O_5$ ($0.01\sim0.09$ for $N_2O_5$ and $6 \times10^{-3}$ for $ClNO_2$), leading to an overall decrease of nitrate production. As the $ClNO_2$ production from the heterogeneous reaction leads to less $N_2O_5$ conversion to non-relative nitrate, it may change the overall

lifetime of NOx and their transport distances. The magnitude of this change and its
implications on ozone and $PM_{2.5}$ in local and downwind areas should be further studied.

3.4 Relationship between nitrate formation and chlorine chemistry
Nitrate productions from the homogeneous and heterogeneous pathways in Beijing are
approximated by the difference in predicted nitrate concentrations between the BASE
or HET case and a sensitivity case without heterogenous reactions. Averaging over the
five-day period, approximately 58% of the nitrate originates from $HNO_3$ gas-to-particle
partitioning and 42% is from heterogeneous reactions (Figure 4). This conclusion
generally agrees with measurements at Peking University (PKU) (52% from the
heterogeneous process and 48% from $HNO_3$ partitioning) on four polluted days
(average in September 2016 reported by Wang et al. (2017). Slightly higher
contributions of the homogeneous pathway in this study is expected because of high
OH concentrations during the day and lower particle surface areas at night.
The nitrate formation from different homogeneous and heterogeneous pathways in
the BASE case and HET case are further studied. Contributions of different gas phase
pathways are determined using the process analysis tool in CMAQ. Contributions of
different heterogeneous pathways are determined using a zero-out method that turns of
one heterogeneous pathway at a time in a series of sensitivity simulations. Figure 4
shows that the reaction of OH and $NO_2$ is always the major pathway for the formation
of nitrate through homogeneous formation of $HNO_3$ and gas-to-particle partitioning.
However, its nitrate production rate through this homogeneous pathway decreases
significantly from daytime to nighttime (from 1.81 µg m$^{-3}$ h$^{-1}$ to 0.33 µg m$^{-3}$ h$^{-1}$ on
average). The nitrate production from other $HNO_3$ partitioning pathways in the daytime
is negligible. At nighttime, homogeneous reaction of $N_2O_5$ with water vapor accounts
for approximately 5% of the overall homogeneous nitrate formation. For the
heterogeneous pathways, daytime production rate is approximately 0.6 µg m$^{-3}$ h$^{-1}$ with
1/3 of the contributions from $NO_2$ and 2/3 from $N_2O_5$. Nighttime production on nitrate
from the heterogeneous pathways is approximately 3.1 µg m$^{-3}$ h$^{-1}$, of which 85% is due
to $N_2O_5$ and 15% is due to $NO_2$.
Comparing the BASE case and the HET case shows that, when the chlorine
chemistry is included, the gaseous $HNO_3$ produced by OH reacting with $NO_2$ increases
significantly in the HET case. Correspondingly, the nitrate production rate reaches up
to 2.04 µg m$^{-3}$ h$^{-1}$ in the daytime due to increased atmospheric OH concentrations
predicted by the chlorine reactions. Similar conclusions are also obtained by Li et al.
(2016) and Liu et al. (2017) based on observations and model simulations. The
heterogeneous production of nitrate from the reaction of $N_2O_5$ uptake decreases by
approximately 27% in the HET case due to the production of gas phase $ClNO_2$.
According to the study by Sarwar et al. (2012; 2014), including the heterogeneous
reaction of $N_2O_5$ with PCl decreased the nocturnal nitrate concentration by 11-21% in
the United States, which was slightly less than the current study for Beijing. It is likely
because PCl concentrations in the United States are significantly lower than those in
Beijing (the monthly PCl concentration is 0.06 µg m$^{-3}$ in the United State against ~1 µg
m$^{-3}$ in Beijing) so that PCl is depleted quickly. The contributions of $NO_2$ uptake to
nitrate also decrease by 22% because of the lower rate constant of the reaction of $NO_2$
with PCl. In contrast, the contribution of $ClNO_2$ reacts with particle surface to nitrate
production increases by 6% in the HET case. The overall nitrate concentration in the
HET case is about 22% higher than that in the BASE case during this study period.

## 4. Conclusions

In this work, a modified CMAQ model incorporated with heterogeneous reactions for
the production of molecular chlorine and other reactive chlorine species is developed
and its impact on of the nitrate formation predictions are evaluated. The contributions
from different homogenous and heterogeneous pathways to nitrate formation are also
quantified. High concentration of $Cl_2$ and $ClNO_2$ do not occur in the southern part of
the Beijing-Tianjin-Hebei region with intensive emissions of chlorine species as higher
concentrations of $O_3$ and $N_2O_5$ associated with the heterogeneous formation of these
species generally occurred in the downwind areas. CTMs without a complete treatment
of the chlorine chemistry can underestimate daytime nitrate formation from the
homogeneous pathways, particularly from $HNO_3$ gas-to-particle partitioning due to

underestimation of OH concentrations and overestimate the nighttime nitrate formation from the heterogeneous pathways due to missing chlorine heterogeneous chemistry.

*Data availability*. The data in this study are available from the authors upon request (shxwang@tsinghua.edu.cn)

*Author contributions*. XQ, QY, SW and JH designed the study; YS, BL, AS, XY provided observation data; XQ, QY, SW, JZ, QX, DD, LD and JX analyzed data. XQ, QY and SW wrote the paper.

*Competing interests*. The authors declare that they have no conflict of interest.

*Acknowledgments.* This work was supported by National Natural Science Foundation of China (21625701), China Postdoctoral Science Foundation (2018M641385), National Research Program for Key Issue in Air Pollution Control (DQGG0301, DQGG0501) and National Key R&D Program of China (2018YFC0213805, 2018YFC0214006). The simulations were completed on the "Explorer 100" cluster system of Tsinghua National Laboratory for Information Science and Technology.

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

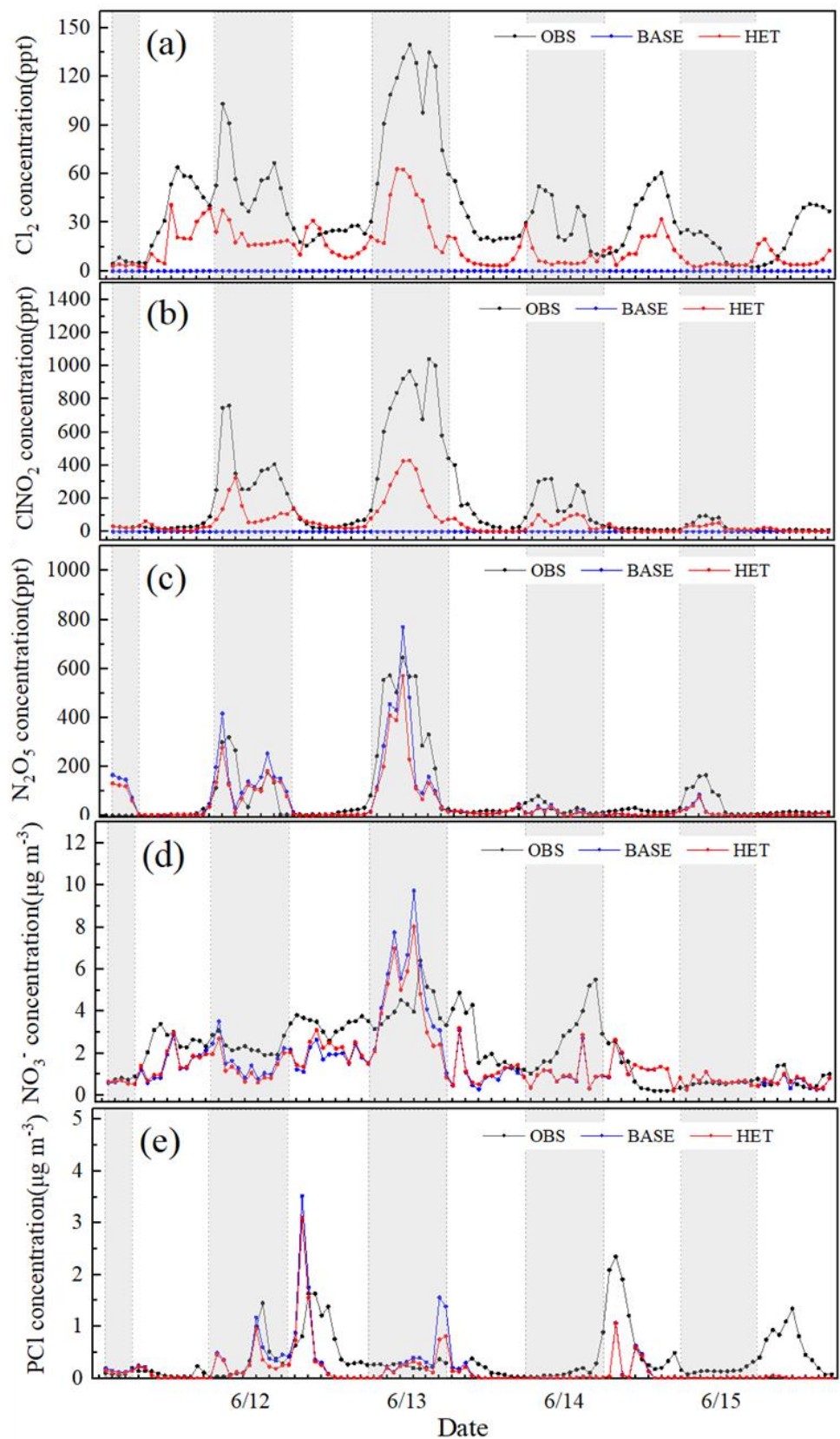

Figure 1 Comparison of observed hourly Cl$_2$, ClNO$_2$, N$_2$O$_5$ (at the Institute of
Atmospheric Physics, Chinese Academy of Sciences), NO$_3^-$ and PCl (at Tsinghua
University) in urban Beijing with predictions from the BASE and the HET cases
during 11-15 June 2017.

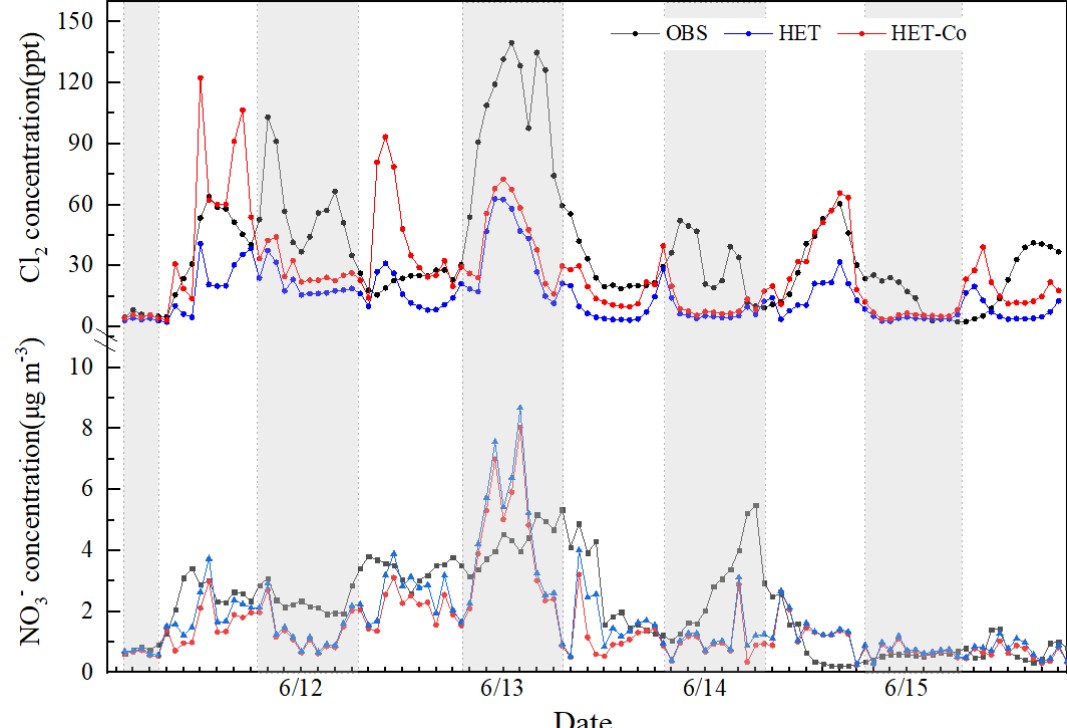

Figure 2 Comparison of observed and predicted $Cl_2$ and $NO_3^-$ concentrations under
different uptake coefficient of $O_3$ (HET: daytime $\gamma_{O_3} = 1 \times 10^{-3}$, nighttime $\gamma_{O_3} = 1 \times 10^{-5}$; HET-Co: daytime $\gamma_{O_3} = 1 \times 10^{-2}$, nighttime $\gamma_{O_3} = 1 \times 10^{-4}$).
$1 \times 10^{-5}$; HET-Co: daytime $\gamma_{O_3} = 1 \times 10^{-2}$, nighttime $\gamma_{O_3} = 1 \times 10^{-4}$).

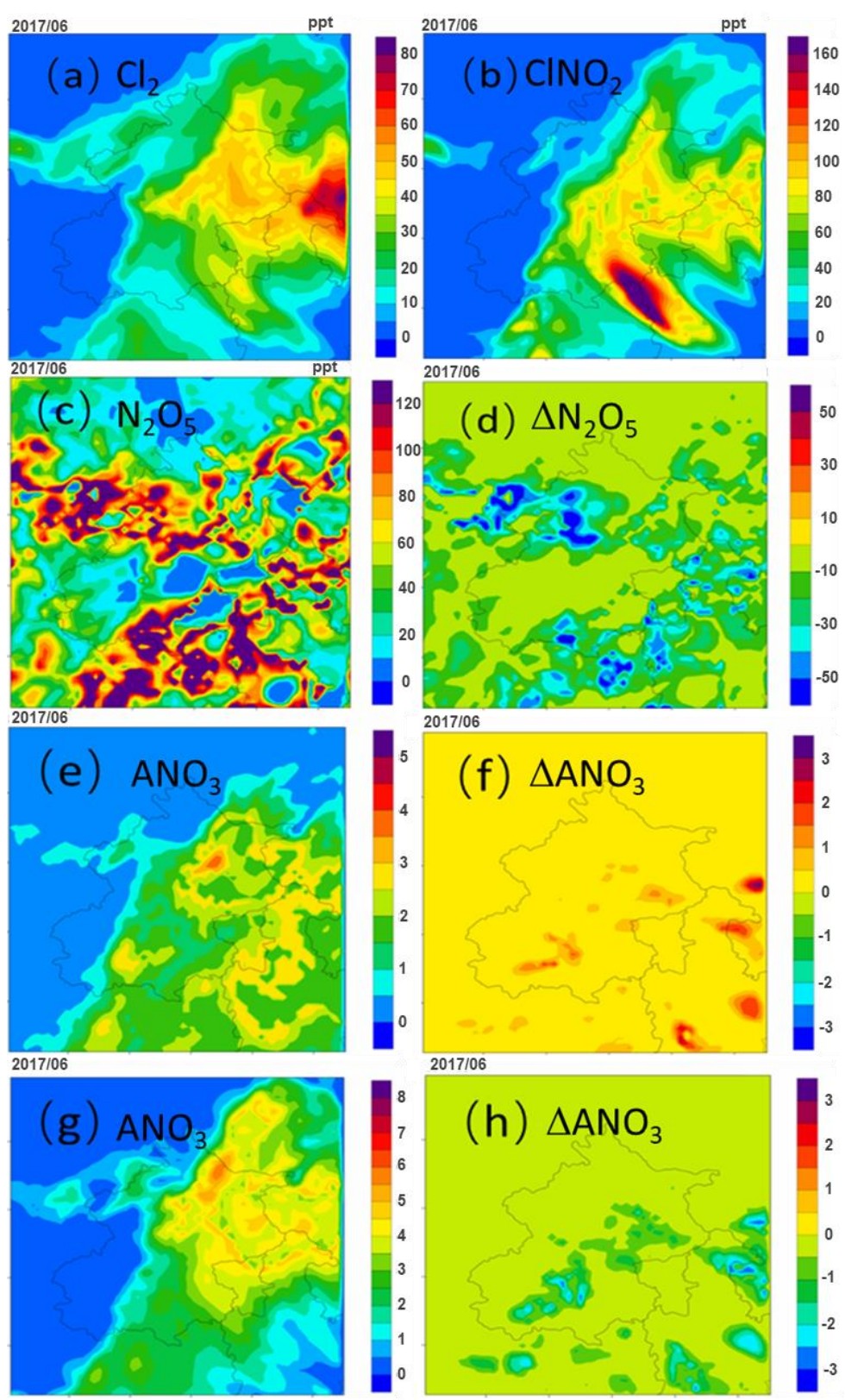

Figure 3 Spatial distributions of episode-average (a) $Cl_2$, (b) $ClNO_2$, (c) $N_2O_5$, (e)
daytime nitrate ($ANO_3$) and (g) nighttime nitrate concentrations from 11-15 June 2017,
and the differences in the episode-average (d) $N_2O_5$ (HET case – BASE case), (f)
daytime nitrate and (g) nighttime nitrate. Units are $\mu g\ m^{-3}$.

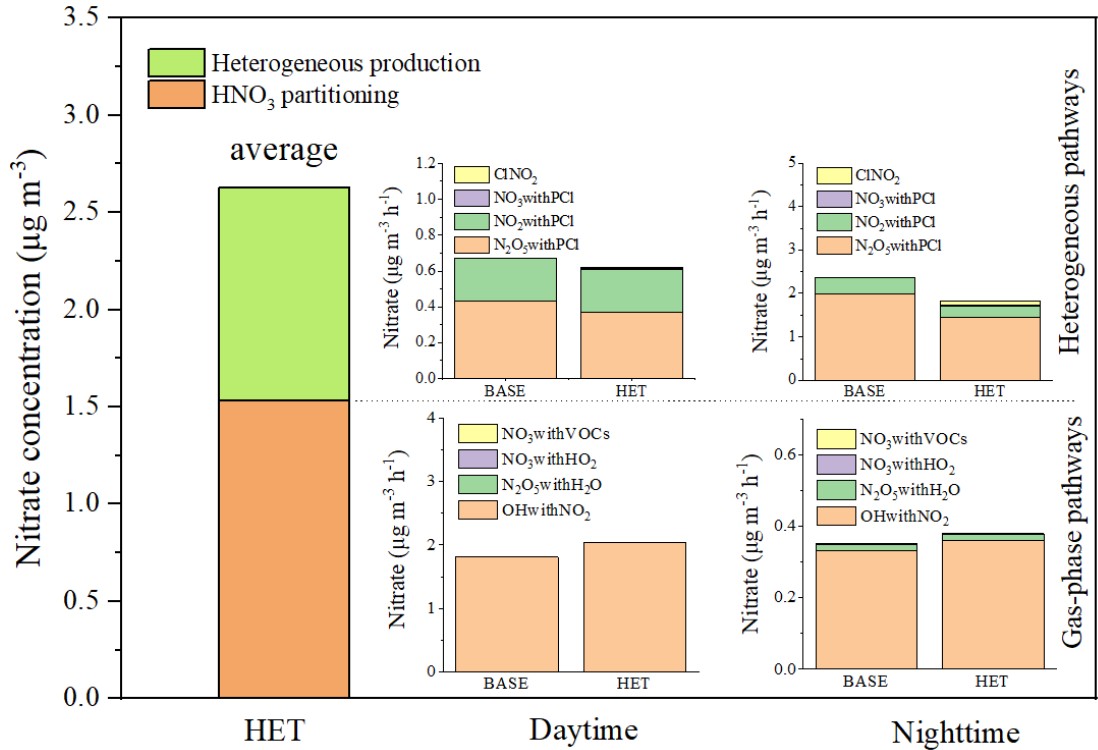


Figure 4 Contributions of different homogeneous and heterogeneous pathways to
nitrate formation.

Table 1 The sectoral emissions of HCl, $Cl_2$ and PCl in Beijing in 2017. Unit: Mg year$^{-1}$

| Sector | Emissions | | |
|---|---|---|---|
| | HCl | $Cl_2$ | PCl |
| Power plant | 22.8 | 1.2 | 6.75 |
| Industry | 587.3 | 20.1 | 89.2 |
| Residential | 202.4 | 8.1 | 34.7 |
| Biomass burning | 0.182 | 0 | 0.14 |
| Municipal solid waste | 1080.2 | 0 | 8.47 |
| Cooking | 0 | 0 | 426.8 |
| Total | 1892.9 | 29.4 | 566.1 |



Table 2 Major gas-phase and heterogeneous pathway of producing nitrate in original CMAQ and newly added or revised heterogeneous reactions
in improved CMAQ.

| Type | Reactions | No. | Reference | Comment |
|---|---|---|---|---|
| **Original CMAQ** | | | | |
| Gas-phase chemistry | $OH + NO_2 \rightarrow HNO_3$ | R1 | | |
| | $N_2O_5 + H_2O \rightarrow 2HNO_3$ | R7 | | |
| | $HO_2{\cdot} + NO_3 \rightarrow 0.2HNO_3 + 0.8OH{\cdot} + 0.8NO_2$ | R8 | | |
| | $NO_3 + VOCs^a \rightarrow HNO_3$ | R9 | | |
| Heterogeneous chemistry | $N_2O_5(g) + H_2O(aq) \rightarrow 2H^+ + 2NO_3^-$ | R5 | | |
| | $2NO_2(g) + H_2O(aq) \rightarrow HONO(g) + H^+ + NO_3^-$ | R10 | | |
| **Improved CMAQ** | | | | |
| Newly added or revised heterogeneous reactions | $N_2O_5(g) + H_2O(aq) + Cl^-(aq) \rightarrow ClNO_2(g) + NO_3^-$ | R6 | Bertram and Thornton (2009) | Revise R5 |
| | $2NO_2(g) + Cl^- \rightarrow ClNO(g) + NO_3^-$ | R11 | Abbatt et al. (1998) | Revise R10 |
| | $NO_3(g) + 2Cl^- \rightarrow Cl_2(g) + NO_3^-$ | R12 | Rudich et al. (1996) | Increase $NO_3^-$ |
| | $O_3(g) + 2Cl^- + H_2O(aq) \rightarrow Cl_2(g) + O_2(g) + 2OH^-$ | R13 | Abbatt et al. (1998) | Affect OH |
| | $2OH{\cdot}(g) + 2Cl^- \rightarrow Cl_2(g) + 2OH^-$ | R14 | George et al. (2010) | Affect OH |
| | $ClONO_2(g) + Cl^- \rightarrow Cl_2(g) + NO_3^-$ | R15 | Deiber et al. (2004) | Affect OH |
| | $HOCl(g) + Cl^- + H^+ \rightarrow Cl_2(g) + H_2O$ | R16 | Pratte et al. (2006) | Affect OH |
| | $ClNO_2(g) + Cl^- + H^+ \rightarrow Cl_2(g) + HONO(g)\ (pH < 2.0)$ | R17 | Riedel et al. (2012) | Affect OH |
| | $ClNO_2(g) + H_2O(aq) \rightarrow Cl^- + NO_3^- + 2H^+\ (pH \geq 2.0)$ | R18 | Rossi (2003) | Increase $NO_3^-$ |

[a]: presents different VOCs species. In the SAPRC-11 mechanism, the VOCs species include CCHO (Acetaldehyde), RCHO (Lumped C3+
Aldehydes), GLY (Glyoxal), MGLY (Methyl Glyoxal), PHEN (phenols), BALD (Aromatic aldehydes), MACR (Methacrolein), IPRD (Lumped
isoprene product species).

Table 3 Observed day (D) and night (N) $NO_3^-$ concentrations (Obs.) and predicted uptake coefficient of $N_2O_5$ ($\gamma_{N2O5}$) and nitrate concentrations (Pred.) using the parameterizations of $\gamma_{N2O5}$ by Bertram and Thornton (2009) (Scenario 1), Davis et al., (2008) (Scenario 2) and the upper-limit value derived by Zhou et al. (2018) (Scenario 3)

| | $NO_3^-$ Obs. | Scenario1 | | Scenario2 | | Scenario3 | |
|---|---|---|---|---|---|---|---|
| | | $\gamma_{N2O5}$ | Pred. | $\gamma_{N2O5}$ | Pred. | $\gamma_{N2O5}$ | Pred. |
| 06/11-D | 2.54 | 0.033 | 1.59 | 0.008 | 1.32 | 0.09 | 2.17 |
| 06/11-12-N | 2.42 | 0.043 | 1.67 | 0.037 | 1.37 | 0.09 | 2.12 |
| 06/12-D | 3.39 | 0.028 | 2.16 | 0.032 | 2.74 | 0.09 | 3.13 |
| 06/12-13-N | 4.24 | 0.021 | 4.02 | 0.022 | 4.05 | 0.09 | 6.04 |
| 06/13-D | 2.57 | 0.012 | 1.18 | 0.008 | 1.06 | 0.09 | 2.47 |
| 06/13-14-N | 4.10 | 0.022 | 4.45 | 0.022 | 4.45 | 0.09 | 7.13 |
| 06/14-D | 0.95 | 0.001 | 1.34 | 0.001 | 1.33 | 0.09 | 1.64 |
| 06/14-15-N | 2.75 | 0.013 | 1.00 | 0.007 | 0.96 | 0.09 | 2.33 |
| 06/15-D | 0.75 | 0.001 | 0.66 | 0.001 | 0.66 | 0.09 | 1.11 |