# Peer review of "Modeling the impact of heterogeneous reactions of chlorine on summertime nitrate formation in Beijing, China"

_Atmospheric Chemistry and Physics, 2018_

## Referee Comment (RC1) · Anonymous Referee #1 · 8 Feb 2019

General comment:

Qiu et al. further developed a widely-used regional chemical transport model, CMAQ, to include several heterogeneous reactions related to chlorine species and applied the revised model in Beijing to estimate the effect of these heterogeneous reactions on the formation of nitrate aerosol in the summertime.

The paper is generally well written and has the potential to contribute to the growing body of the studies on tropospheric halogen chemistry and its impact on air quality. However, there are several major issues and some minor comments that should be addressed before it can be accepted for the publication in Atmospheric Chemistry and

[Figure]

Physics.

One of the major concerns is that the authors omitted several important papers related to chlorine and nitrogen chemistry, e.g., Brown and Stutz (2012), Osthoff et al. (2008), Sarwar et al. (2012), and Sarwar et al. (2014). These papers should be included in Section 1 (Introduction and Research background), in Section 2.2 (model development), or Section 3.3 and 3.4 (model results and discussion). See the specific comments below.

The second major issue is that the current manuscript does not include any information related to NO2, O3, and PM2.5, which are the precursors of N2O5, ClNO2, and nitrate. No emission of these pollutants is described. No model evaluation. No simulation results. Without this information, it is difficult to assess the model performance and therefore the outcome of the simulation.

The last main problem is that there are too many errors and typos throughout the manuscript, e.g., citing the improper reference, citing the reference that is not in the reference list, the reference list is not organized according to the alphabet, wrong spelling, etc. Please refer to the technical comments. I suggest that the authors carefully read through and thoroughly revise their manuscript.

Specific comment:

1.Line 26-28. These descriptions are redundant to line 33-36.

2.Line 37-39. The ClNO2 production decreases nitrate during nighttime and increases nitrate during the daytime. Does it mean that the chlorine chemistry changes the temporal pattern of the nitrate formation and therefore the spatial pattern? Does it have any implication to the air quality control? I would love to see a discussion on this implication.

3.Line 50-57. The authors only introduced two production pathways of the secondary nitrate. However, the other pathways, e.g., those in Table 2, also play non-negligible

roles. Should add those pathways in the introduction.

4. Line 54. A reference is needed for the 'enhancement effect of NH3-NH4+ gas-particle equilibrium on the nitrate formation'.

5. Line 57. These papers are not the proper reference for the nitrate formation mechanism, e.g., Brown and Stutz (2012) is a better one for the N2O5 (NO3) chemistry.

6. Line 63-72. The authors only introduced three previous works here, and all of them were conducted in China, in the Northern China Plain to be exact. What about similar modeling studies in other regions, e.g., the southern part of China, Northern America, and Europe? For example, Sarwar et al. (2012, 2014) developed the same model, CMAQ, to evaluate the effect of ClNO2 production on air quality, including the total nitrate, in the US and the Northern Hemisphere. However, these two critical papers are not discussed anywhere in the current manuscript.

7. Line 73. This statement might be true, but the authors did not provide any evidence/reference to support it.

8. Line 77-80. This statement is not correct. For example, Wang et al. (2016) and Brown et al. (2016) reported extremely high N2O5 mixing ratios at a site in Hong Kong (a coastal city) of up to 8 ppbv (1 min average) or 12 ppbv (1 s average). This brings up another issue. Should include the average time when report observational results, e.g., 1 s average, 1 min average, or 1h average.

9. Line 79-80. There is no Li et al. (2017) in the reference list. Are you referring to Li et al. (2016)? That is not a proper reference here, because that paper is a modeling study that used the measurement results from Wang et al. (2016).

10. Line 82. These references are not the right ones here. The first measurements of ClNO2 in the real atmosphere, Osthoff et al. (2008) and Thornton et al. (2010), are better ones.

11. Line 102. This is not entirely true. For instance, Hossaini et al. (2016) developed a

global chemical transport model (TOMCAT) and included several heterogeneous reactions of chlorine species on chloride-containing aerosol.

12.Line 107-111. I assume the replacement is only in Beijing city but not in the surrounding areas. Is that correct?

13.Line 117-118. This sentence is a bit out-of-blue. The following sentence makes more sense to me.

'Thus, it is important to compile an updated emission inventory for Beijing (and its surrounding areas) to include the emissions from cooking and other sources (coal burning, solid waste burning, biomass burning, etc.)."

14.Line 128-130. Should add some reference here. Also, 'NH3' should also be considered as a common species.

15.Line 136. Should mention the number (from 2000 Mt in 2014 to 490 Mt in 2017) here.

16.Line 142. Should list the emission factors for different sectors, at least in the supplement. Also, give the reference.

17.Line 156-157. Should provide reasons why you chose this number of hours. Three hours of cooking time seem to be a bit long for me. Also, 'restaurant' should be 'social cooking', is that correct?

18.Line 160. Any reason that you chose '150'?

19.Line 173-174. A brief description is needed for other emissions, which are the precursors of N2O5, ClNO2, and nitrate aerosol.

20.Section 2.2. The authors added several reactions to the CMAQ model, and this seems to be one of the major contributions of this study. However, what is the difference between the mechanism in the current study and that in Sarwar et al. (2012, 2014)? I notice that one of the co-authors in the present study is also a co-author of Sarwar

et al. (2014). I strongly advise the authors to carefully review the previous works and identify the advantage of the current work, instead of avoiding the comparison between the current study with the previous ones.

Besides, did you compare your scheme with Zheng et al. (2015)?

21.Line 178. What do you mean by 'current CMAQ model'? Is 'Zheng et al. (2015)' a proper reference for 'current CMAQ model'?

22.Line 182. Li et al. (2016) is not a proper reference for this equation. Should refer to Bertram and Thornton (2009) or Roberts et al. (2009) with a very similar formula. This brings up another persistent issue that you should use 'Bertram and Thornton (2009)' instead of 'Bertram et al. (2009)'. There are only two authors to that paper. Please check the manuscript for this error.

23.Line 177. $H_2O$ means water vapor. Is that right?

24.Line 220-22. This part is a bit confusing. Why did you call the values 'preliminary'? Did you get these data from the reference (Keene et al., 1999)? Or did you make a guess on these values? I see that you made some sensitivity cases later. Perhaps you should mention that here.

25.Line 229. Just out of curiosity, is the gas-phase chemistry of chlorine the same as that in Sarwar et al. (2012)?

26.Section 3.1. What about the model performance on the simulation of $NO_2$, $O_3$, and $PM2.5$? These are particularly important to the formation of nitrate aerosol, the sole focus of the current work.

27.Line 251-252. This treatment artificially amplifies the heterogeneous reaction rate by 5 and 10 times. Is it logical to do that based on the comparison with one measurement data set? How does the CMAQ model perform in general in the simulation of surface area? Is it a common issue? Or did it only happen in your simulation? How is the simulation of $PM2.5$? This highlights the need for the evaluation of the routine air

pollutants, e.g., PM2.5.

28.Line 255. ClNO2 is underestimated in your cases. Does it mean that nitrate aerosol is over-predicted?

29.Line 258. The O3 will increase or decrease if you change the uptake coefficient of O3. This also calls for the model evaluation on O3.

30.Line 260. Several studies have been reported that the parameterization used in the present study (Bertram and Thornton, 2009) has some uncertainty, including Tham et al. (2018), McDuffie et al. (2018a), McDuffie et al. (2018b) and the references therein. How these uncertainties affect the conclusions should be discussed.Âă

31.Line 277-278. The underestimated nitrate concentration could be due to many reasons. For example, is NO2 underestimated as well? This is another reason to show the evaluation of NO2 simulation. The uncertainty of the parameterizations of N2O5 uptake and ClNO2 yield (comment 31) could also be applied here. Besides, how did you simulate the most critical OH precursor, i.e., HONO? Did you evaluate your HONO simulation? Did you consider the NO2 uptake on environmental surfaces? What about other HONO sources? Several recent papers have shown that HONO is very important in simulating nitrate, e.g., Fu et al. (2018).

32.Line 281-282. What do you mean by 'the nitrate level is higher in the daytime and lower in nighttime'?

33.Line 290-291. It may be true that the uptake coefficients for these two molecules are the most important parameters. But what about the uptake coefficients for other species/process. Did you run any sensitivity cases to examine that?

34.Line 293-295. Or maybe just because the direct emission of Cl2 is not well represented in the emission inventory.

35.Line 316. N2O5 uptake process is very complicated. The word, 'inferior', is definitely not the one I would use to describe the parameterization based on nitrate and sulfate

concentration. Please revise. The uncertainty of the parameterizations of N2O5 uptake and ClNO2 yield also works here.

36.Section 3.3. I advise to include the simulated results of NO2, O3, and PM2.5.

37.Line 328. Should include a figure for the 'intensive emissions of chlorine species'.

38.Line 328-330. The presence of a high concentration of ClNO2 and Cl2 away from the fresh emissions does not necessarily mean that ClNO2 and Cl2 are easy to transport.

For example, the production of ClNO2 requires the presence of chloride, NO2, and O3. In the areas close to the fresh emissions, O3 is commonly low, and the production of NO3 (hence N2O5 and ClNO2) is limited. Therefore, the production of ClNO2 is generally not found near fresh emissions.

As to the Cl2, perhaps the contribution of direct emission to the level of Cl2 is not significant, and Cl2 is predominantly produced in the atmosphere. So the high levels of Cl2 are found away from the fresh emissions.

39.Line 336. Why 'more N2O5 is converted into nitrate'? Are you implying that the uptake coefficient calculated with the Bertram and Thornton (2009) is higher than that with Davis et al. (2008)?

40.Line 351-352. Was the observation in PKU conducted in the same period?

41.Line 355-357. Are you implying that in cleaner days, the OH level is higher, so the production of HNO3 from OH + NO2 is more important?

42.Line 371-372. How did you treat the reaction of NO2 + H2O (aq)? Did you revise it to NO2+ Cl- or did you use both?

43.Line 396-397. It is correct that the emission of chlorine species is vital to chlorine chemistry study. But the current study does not demonstrate this point.
44.Line 402-404. Not necessarily correct. See comment above.

45.Line 409-411. What implications? Care to elaborate? See comment 2 for example. But more thoughts are definitely of value to the policymakers.

46. The reference list is not organized according to the alphabet. For example, L is before K, J is after K, Rudich is before Roberts, and Spicer is before Song.

47.Table 2. What are the effects of R6, R11, R13-R18 on the production of nitrate aerosol? Also, please provide the reference for all reactions.

Technical comment:

48.Line 38. 'photolyze' is a better word than 'decompose' since it is a photolysis reaction.

49.Line 62. Should add '(aq)' after 'H2O'.

50. Line 63. should define CMAQ and WRF-Chem here.

51. Line 81. should be 'chloride-containing', not 'chlorine-containing'.

52.Line 119. The definition of CMAQ should be moved to line 63.

53.Line 130. Add a space between 'emissions' and 'were'.

54. Line 145. 'EF represents the emission factor' should be 'EFi,j represents the emission factor of pollutant j in sector i'.

55.Line 148-149. 'had been detailed described' should be 'had been described in detail'.

56.Line 158. Add 'from social cooking' after 'Vc is the volume of exhaust gas'.

57.Line 161. 'chose' should be 'chosen'.

58.Line 166. Delete 'that'.

59.Line 168. Use the same decimal for all data.

60.Line 169. 'Others' should be 'Other'.

61.Line 169. Add the publication year after 'Fu et al.'

62.Line 169 and line 171. Two 'finally'?

63.Line 185. 'Laboratorial' should be 'laboratory'.

64.Line 192. Do you mean 'equation (5)', instead of 'equations (2)'?

65.Line 198. Do you mean 'equation (6)', instead of 'equation (3)'?

66. Why did you use different terms for velocity in equation 5 and equation 8?

67.Line 211. How did you calculate Kh, Kf, K3/K2, and K4/K2? Are they constants? If so, please add the number.

68.Line 216 and line 208. Use the consistent form for units. m3/m3 or m3 m-3.

69.Line 227. Which year?

70.Line 232. It is weird to see 'Figure 3' before 'Figure 1 and 2'. Maybe a map with three domains in the supplement as Figure S1 is better.

71.Line 232. '40oE' should be '40oN'.

72.Line 296. Remove the extra space between 'empirical' and 'and'.

73.Line 315. Add the year for 'Davis et al.'. Check through the manuscript for a similar issue.

74.Line 350. 'are produced' should be 'is produced'.

75.Line 369. Add 'of' between 'uptake' and 'N2O5'.

76.Line 378. 'even through' should be 'even though'.

77.Line 396. 'This' should be 'These'.

78.Line 397. 'becase' should be 'because'.

79.Line 397. 'the cornerstones' should be 'the cornerstone' or 'one of the cornerstones'.

80.Line 401. 'chlorine' should be 'chloride'.

81.Line 409. 'understnadings' should be 'understandings'.

82.Figure 3. Should point out the area of BJ and the location of the sampling site. In d, f, and h, should use △N2O5, △NO3-, and △NO3-, instead of N2O5, NO3-, and NO3-

83.Figure 4. In the sub-plot Daytime Gas-phase, the title of Y-axis should be 'HNO3 production rate (ppt h-1)'. The same revision should be applied to the sub-plot Nighttime Gas-phase. The sub-plot Nighttime Heterogeneous, the title of Y-axis should be 'Nitrate production rate ($\mu$g m-3 h-1)'. No sub-plot daytime Heterogeneous?

Reference:

Brown, S.S. and Stutz, J., 2012. Nighttime radical observations and chemistry. Chemical Society Reviews, 41(19), pp.6405-6447.

Brown, S.S., Dubé, W.P., Tham, Y.J., Zha, Q., Xue, L., Poon, S., Wang, Z., Blake, D.R., Tsui, W., Parrish, D.D. and Wang, T., 2016. Nighttime chemistry at a high altitude site above Hong Kong. Journal of Geophysical Research: Atmospheres, 121(5), pp.2457-2475.

Fu, X., Wang, T., Zhang, L., Li, Q., Wang, Z., Xia, M., Yun, H., Wang, W., Yu, C., Yue, D. and Zhou, Y., 2019. The significant contribution of HONO to secondary pollutants during a severe winter pollution event in southern China. Atmospheric Chemistry and Physics, 19(1), pp.1-14.

Hossaini, R., Chipperfield, M.P., Saiz‐Lopez, A., Fernandez, R., Monks, S., Feng,

W., Brauer, P. and Glasow, R., 2016. A global model of tropospheric chlorine chemistry: Organic versus inorganic sources and impact on methane oxidation. Journal of Geophysical Research: Atmospheres, 121(23).

McDuffie, E.E., Fibiger, D.L., Dubé, W.P., Lopez‐Hilfiker, F., Lee, B.H., Thornton, J.A., Shah, V., Jaeglé, L., Guo, H., Weber, R.J. and Michael Reeves, J., 2018a. Heterogeneous N2O5 uptake during winter: Aircraft measurements during the 2015 WINTER campaign and critical evaluation of current parameterizations. Journal of Geophysical Research: Atmospheres, 123(8), pp.4345-4372.

McDuffie, E.E., Fibiger, D.L., Dubé, W.P., Lopez Hilfiker, F., Lee, B.H., Jaeglé, L., Guo, H., Weber, R.J., Reeves, J.M., Weinheimer, A.J. and Schroder, J.C., 2018b. ClNO2 yields from aircraft measurements during the 2015 WINTER campaign and critical evaluation of the current parameterization. Journal of Geophysical Research: Atmospheres, 123(22), pp.12-994.

Osthoff, H.D., Roberts, J.M., Ravishankara, A.R., Williams, E.J., Lerner, B.M., Sommariva, R., Bates, T.S., Coffman, D., Quinn, P.K., Dibb, J.E. and Stark, H., 2008. High levels of nitryl chloride in the polluted subtropical marine boundary layer. Nature Geoscience, 1(5), p.324.

Roberts, J.M., Osthoff, H.D., Brown, S.S., Ravishankara, A.R., Coffman, D., Quinn, P. and Bates, T., 2009. Laboratory studies of products of N2O5 uptake on Cl− containing substrates. Geophysical Research Letters, 36(20).

Sarwar, G., Simon, H., Bhave, P. and Yarwood, G., 2012. Examining the impact of heterogeneous nitryl chloride production on air quality across the United States. Atmospheric Chemistry and Physics, 12(14), pp.6455-6473.

Sarwar, G., Simon, H., Xing, J. and Mathur, R., 2014. Importance of tropospheric ClNO2 chemistry across the Northern Hemisphere. Geophysical Research Letters, 41(11), pp.4050-4058.

Simon, H., Kimura, Y., McGaughey, G., Allen, D.T., Brown, S.S., Osthoff, H.D., Roberts, J.M., Byun, D. and Lee, D., 2009. Modeling the impact of ClNO2 on ozone formation in the Houston area. Journal of Geophysical Research: Atmospheres, 114(D7).

Tham, Y.J., Wang, Z., Li, Q., Wang, W., Wang, X., Lu, K., Ma, N., Yan, C., Kecorius, S., Wiedensohler, A. and Zhang, Y., 2018. Heterogeneous N 2 O 5 uptake coefficient and production yield of ClNO 2 in polluted northern China: roles of aerosol water content and chemical composition. Atmospheric Chemistry and Physics, 18(17), pp.13155-13171.

Thornton, J.A., Kercher, J.P., Riedel, T.P., Wagner, N.L., Cozic, J., Holloway, J.S., Dubé, W.P., Wolfe, G.M., Quinn, P.K., Middlebrook, A.M. and Alexander, B., 2010. A large atomic chlorine source inferred from mid-continental reactive nitrogen chemistry. Nature, 464(7286), p.271.

Wang, T., Tham, Y.J., Xue, L., Li, Q., Zha, Q., Wang, Z., Poon, S.C., Dubé, W.P., Blake, D.R., Louie, P.K. and Luk, C.W., 2016. Observations of nitryl chloride and modeling its source and effect on ozone in the planetary boundary layer of southern China. Journal of Geophysical Research: Atmospheres, 121(5), pp.2476-2489.

Zheng, B., Zhang, Q., Zhang, Y., He, K.B., Wang, K., Zheng, G.J., Duan, F.K., Ma, Y.L. and Kimoto, T., 2015. Heterogeneous chemistry: a mechanism missing in current models to explain secondary inorganic aerosol formation during the January 2013 haze episode in North China. Atmospheric Chemistry and Physics (Online), 15(4).

---

## Referee Comment (RC2) · Anonymous Referee #2 · 2 Mar 2019

The manuscript of Qiu et al., reported the influence of chloride chemistry to particulate nitrate formation in the framework of CMAQ. The authors included several chloride reactions into the CMAQ chemical mechanism. This topic is in the scope of ACP, and would benefit the knowledge of the heterogeneous reactions in the formation of particulate nitrate in Beijing, China. While in this paper, the detail of the parameters lacking the foundation and the analysis of model result seems common. Some of the conclusions seems can't obtain from the model result analysis (see major comments). The following comments should be addressed before publishing in ACP.

Major comments:

[Figure]

1. Line 250-252, the treatment of aerosol surface area by time 5 or 10 in this model is unconvinced. Should provide more evidence to support the rationality.

2. The sensitivity tests used an $O_3$ uptake coefficient enlarged by a factor of 10 without any reference, while the $Cl_2$ simulations do not significantly improved in general. Other possible $Cl_2$ formation channel shall be tested or at least discussed.

3. Line 398-402, I cannot agree with that the parameterization method including chloride of the uptake coefficient of N2O5 has a better performance, at least this kind of conclusion cannot be deduced from the authors analysis (cf. figure 1).

4. I suggest that the authors may present a table to summarize all the revisions of the parameters related to the uptake coefficient as well as the related heterogeneous reactions.

Minor comments:

1. Line 206, Bertram et al., 2009 correct to Bertram and Thornton, (2009).

2. Line 335-line 340, this part is confused. Did you mean more N2O5 convert to nitrate due to the N2O5 uptake coefficient calculated by Bertram and Thornton, (2009) is higher than the base case? And the non-significant nitrate increase may be due to the ClNO2 yield buffered the increasing caused by the application of new N2O5 uptake coefficient?

3. Line 289 the section title should not be the estimation of uptake coefficients of O3 and N2O5, but the influence of the change of these parameters.

---

## Author Comment (AC2) · 14 Apr 2019

**Reviewer #2**

**Comment #1:** The manuscript of Qiu et al., reported the influence of chloride chemistry to particulate nitrate formation in the framework of CMAQ. The authors included several chloride reactions into the CMAQ chemical mechanism. This topic is in the scope of ACP, and would benefit the knowledge of the heterogeneous reactions in the formation of particulate nitrate in Beijing, China. While in this paper, the detail of the parameters lacking the foundation and the analysis of model result seems common. Some of the conclusions seems can't obtain from the model result analysis (see major comments). The following comments should be addressed before publishing in ACP.

**Response #1:** Thanks for the positive comments on this manuscript. The suggestions are addressed in detail in the following.

**Comment #2:** Line 250-252, the treatment of aerosol surface area by time 5 or 10 in this model is unconvinced. Should provide more evidence to support the rationality.

**Response #2:** Figure R1a below is the detailed comparison of our modeled surface area with observations averaged between June 11-15, 2017. Under-prediction of surface area varies between 6-12 times. CMAQ model performance for surface area has not been extensively studied. In the only study we found in the literature (Park et al., 2006), similar under-prediction of particle surface area was reported. In their study, the mass concentration of PM2.5 at Atlanta, United State is generally well reproduced by the CMAQ model but the surface area shows large under-predictions similar to what we have seen in our study (see Figure R1b).

[Figure]

Figure R1a (left panel): Predicted averaged hourly particle wet surface area at IAP (June 11-15, 2017) and the ratio of observed to predicted PSA (O/P). R1b (right panel): CMAO predicted and observed daily particle surface area, volume, and mass concentrations for PM2.5 from 1/1/1999 to 8/31/2000 at the JST station in Atlanta, Georgia. (R1b is adapted from Park et al., 2006).

Reference:
Park, S.K., Marmur, A., Kim, S.B., et al.: Evaluation of fine particle number concentrations in CMAQ. Aerosol Science and Technology, 40, 985-996, 2006.

**Comment #3:** The sensitivity tests used an O3 uptake coefficient enlarged by a factor of 10 without any reference, while the Cl2 simulations do not significantly improved in general. Other possible Cl2 formation channel shall be tested or at least discussed.

**Response #3:** In Keene et al. (1990) [It was incorrectly cited in the original manuscript as Keene et al. (1999). We apologize for this mistake.], the daytime uptake coefficient of $O_3$ was not based on direct experimental measurements but was estimated indirectly based on a steady-state analysis of $Cl_2$ production rate in a hypothesized geochemical cycle of reactive inorganic chlorine in the marine boundary layer using a 0-D box model. Such an analysis tends to have large uncertainties and Keene et al. (1990) proposed $10^{-4}$ to $10^{-3}$. Nighttime value of $10^{-5}$ was proposed without much supporting evidence. We chose to increase these uptake coefficients to explore the upper limit of the impact of $O_3$ uptake on $Cl_2$ formation. As there are additional $Cl_2$ formation pathways, our results show that the higher uptake coefficients alone do not lead to significantly higher $Cl_2$ concentrations. The other Cl2 channels are summarized in Table 2 and contributions of these channels to $Cl_2$ concentrations are explored in a separate study as this paper is focused on nitrate.

**Comment #4:** Line 398-402, I cannot agree with that the parameterization method including chloride of the uptake coefficient of N2O5 has a better performance, at least this kind of conclusion cannot be deduced from the authors analysis (cf. figure 1).

**Response #4:** Agree. This is removed from the conclusion section.

**Comment #5:** I suggest that the authors may present a table to summarize all the revisions of the parameters related to the uptake coefficient as well as the related heterogeneous reactions.

**Response #5:** We summary the revision of parameters in Table S3.

**Comment #6:** Line 206, Bertram et al., 2009 correct to Bertram and Thornton, (2009).

**Response #6:** Revised

**Comment #7:** Line 335-line 340, this part is confused. Did you mean more N2O5 convert to nitrate due to the N2O5 uptake coefficient calculated by Bertram and Thornton, (2009) is higher than the base case? And the non-significant nitrate increase may be due to the ClNO2 yield buffered the increasing caused by the application of new N2O5 uptake coefficient?

**Response #7:** This sentence is revised as 'By incorporating the chlorine heterogeneous reaction, the $N_2O_5$ concentrations decrease by about 16% because the uptake coefficient calculated with the Bertram and Thornton (2009) is higher than that with Davis et al. (2008).'

**Comment #8:** Line 289 the section title should not be the estimation of uptake coefficients of O3 and N2O5, but the influence of the change of these parameters.

**Response #8:** The title is revised as 'Impact of uptake coefficients of $O_3$ and $N_2O_5$ on chlorine species and nitrate'

---

## Author Response (AR1)

**Response to reviewers' comments**

Thanks to the reviewers for giving us very useful comments to improve our manuscript entitled "Modeling the impact of heterogeneous reactions of chlorine on summertime nitrate formation in Beijing, China" **(acp-2018-1270)**. Detailed responses to reviewers' comment are list below:

**Reviewer #1**

**Comment 1:** Qiu et al. further developed a widely-used regional chemical transport model, CMAQ, to include several heterogeneous reactions related to chlorine species and applied the revised model in Beijing to estimate the effect of these heterogeneous reactions on the formation of nitrate aerosol in the summertime. The paper is generally well written and has the potential to contribute to the growing body of the studies on tropospheric halogen chemistry and its impact on air quality. However, there are several major issues and some minor comments that should be addressed before it can be accepted for the publication in Atmospheric Chemistry and Physics.

**Response #1**: Thanks for the positive comments on this manuscript. The suggestions are addressed in detail in the following.

**Comment 2:** One of the major concerns is that the authors omitted several important papers related to chlorine and nitrogen chemistry, e.g., Brown and Stutz (2012), Osthoff et al. (2008), Sarwar et al. (2012), and Sarwar et al. (2014). These papers should be included in Section 1 (Introduction and Research background), in Section 2.2 (model development), or Section 3.3 and 3.4 (model results and discussion). See the specific comments below.

**Response #2**: We have included these important literatures in our revised manuscript, please see the detailed description in Response below.

**Comment 3:** The second major issue is that the current manuscript does not include any information related to $NO_2$, $O_3$, and $PM_{2.5}$, which are the precursors of $N_2O_5$, $ClNO_2$, and nitrate. No emission of these pollutants is described. No model evaluation. No simulation results. Without this information, it is difficult to assess the model performance and therefore the outcome of the simulation.

**Response #3:** Emissions of conventional species, including $SO_2$, NOx, VOCs, PM2.5 and PM10 for anthropogenic sectors for this study period has been developed using tools developed in our group and the method has been described in previous studies (Wang et al., 2014). For this study period, emissions are generated using the same system with updated input data for activities, controls, emission factors and speciation factors for 2017. More details of the emission processing processes are documented by Ding et al. (under review). A summary of the annual emissions in Beijing in 2017 based on Ding et al. is included in Table S1 in the revised manuscript.

Evaluations of predicted $O_3$, $NO_2$ and $PM_{2.5}$ concentrations are now described in the revised manuscript on page 34, lines 325-328: "Predicted $O_3$, $NO_2$ and $PM_{2.5}$ concentrations from the BASE case simulation are evaluated against monitoring data at 12 monitoring sites in Beijing (Table S2) for 11 to 15 June 2017. The average NMB/NME values for $O_3$, $NO_2$ and $PM_{2.5}$ across the 12 sites are -8%/29%, -7%/59% and -8%/53%, respectively.". Table S2 is attached below as Table R1 for the convenience of the reviewer.

Table R1 Comparison of simulated episode average hourly $NO_2$ and $PM_{2.5}$ and $O_3$ concentrations with observations averaged from 11 to 15 June 2017 (Obs.: observation, Sim.: simulation). Units: $\mu g\ m^{-3}$

| | $NO_2$ | | | | $O_3$ | | | | $PM_{2.5}$ | | | |
| Sites | Obs. | Sim. | NMB | NME | Obs. | Sim. | NMB | NME | Obs. | Sim. | NMB | NME |
|---|---|---|---|---|---|---|---|---|---|---|---|---|
| WSXG | 49 | 54 | 11% | 55% | 99 | 122 | 23% | 31% | 40 | 38 | -6% | 53% |
| DL | 21 | 17 | -20% | 68% | 111 | 108 | -2% | 12% | 32 | 29 | -10% | 52% |
| DS | 47 | 53 | 13% | 54% | 100 | 114 | 15% | 28% | 44 | 41 | -7% | 53% |
| TT | 40 | 48 | 20% | 64% | 98 | 130 | 33% | 45% | 37 | 37 | 1% | 58% |
| NZG | 51 | 66 | 28% | 62% | 111 | 121 | 9% | 25% | 42 | 39 | -7% | 52% |
| GY | 55 | 65 | 17% | 57% | 107 | 116 | 9% | 22% | 36 | 33 | -8% | 54% |
| WL | 52 | 41 | -21% | 54% | 92 | 112 | 22% | 43% | 35 | 33 | -7% | 54% |
| XC | 43 | 31 | -28% | 47% | 100 | 108 | 8% | 12% | 33 | 29 | -12% | 55% |
| HR | 26 | 11 | -56% | 70% | 124 | 105 | -15% | 27% | 27 | 22 | -19% | 51% |
| CP | 42 | 28 | -34% | 58% | 96 | 91 | -5% | 27% | 33 | 32 | -1% | 54% |
| ATZX | 56 | 62 | 10% | 55% | 105 | 107 | 1% | 18% | 33 | 31 | -4% | 54% |
| GC | 56 | 42 | -25% | 58% | 106 | 107 | 0% | 19% | 43 | 37 | -14% | 52% |

WSXG: Wanshouxigong; DL: Dingling; DS: Dongsi; TT:Tiantan; NZG:Nongzhanguan; GY: Guanyuan; WL: Wanliu; XC:Xincheng; HR:Huairou; CP:Changping; ATZX:Aotizhongxin; GC:Gucheng; NMB: normalized mean bias; NME: normalized mean error.

**Comment #4:** The last main problem is that there are too many errors and typos throughout the manuscript, e.g., citing the improper reference, citing the reference that is not in the reference list, the reference list is not organized according to the alphabet, wrong spelling, etc. Please refer to the technical comments. I suggest that the authors carefully read through and thoroughly revise their manuscript.

**Response #4:** We revise these errors following the reviewer's comments below and fixed errors and typos throughout the manuscript as much as we can.

**Comment #5:** Line 26-28. These descriptions are redundant to line 33-36.

**Response #5:** Thank you for pointing out the redundant descriptions. The sentences

"The results show that these heterogeneous reactions significant increase the atmospheric $Cl_2$ and $ClNO_2$ level, leading to an increase of the nitrate concentration by ~10% in the daytime. However, these reactions also lead to a decrease the nocturnal nitrate by ~20%." in line 25-26 are revised as "The results show that these heterogeneous reactions increase the atmospheric $Cl_2$ and $ClNO_2$ level (~100%), which further affect the nitrate formation"

**Comment #6:** Line 37-39. The $ClNO_2$ production decreases nitrate during nighttime and increases nitrate during the daytime. Does it mean that the chlorine chemistry changes the temporal pattern of the nitrate formation and therefore the spatial pattern? Does it have any implication to the air quality control? I would love to see a discussion on this implication.

**Response #6:** While it is true that temporal pattern of nitrate formation was slightly altered, the spatial patterns of nitrate didn't change significantly during the study period. However, as the $ClNO_2$ production from the heterogeneous reaction leads to less $N_2O_5$ conversion to non-relative nitrate, it may change the overall lifetime of NOx and their transport distances. The magnitude of this change and its implications on ozone and PM2.5 locally and in the downwind areas should be further studied. We included this in the revised manuscript on page 38, lines 473-477.

**Comment #7:** Line 50-57. The authors only introduced two production pathways of the secondary nitrate. However, the other pathways, e.g., those in Table 2, also play non-negligible roles. Should add those pathways in the introduction.

**Response #7:** The other gaseous reactions such as $NO_3 + HO_2$, $VOC + NO_3$, and $N_2O_5$ with water vapor are generally negligible in terms of secondary nitrate formation in polluted urban and rural areas, due to low concentrations of NO3 and HO2, and low yield of HNO3 in the VOC + NO3 reactions. We included them in Table 2 for completeness but we don't think they should be specifically mentioned in the introduction section. The heterogeneous reaction of $NO_2$ could be important so we included a sentence in the revised manuscript to mention that:

"The heterogeneous reaction of $NO_2$ on particle surface has been shown to be an important source of secondary nitrate" (Abbatt et al., 1998).

**Comment #8:** A reference is needed for the enhancement effect of $NH_3$-$NH_4^+$ gas-particle equilibrium on the nitrate formation.

**Response #8:** The reference below are added.

Kleeman, M.J., Ying, Q., Kaduwela, A., 2005. Control strategies for the reduction of airborne particulate nitrate in California's San Joaquin Valley. Atmospheric Environment 39, 5325-5341.

Seinfeld, J.H., Pandis, S.N., 2006. Atmospheric Chemistry and Physics: From Air
Pollution to Climate Change. Wiley-Interscience, New York.

**Comment #9:** Line 57. These papers are not the proper reference for the nitrate
formation mechanism, e.g., Brown and Stutz (2012) is a better one for the $N_2O_5$ ($NO_3$)
chemistry.

**Response #9:** Revised.

**Comment #10:** Line 63-72. The authors only introduced three previous works here,
and all of them were conducted in China, in the Northern China Plain to be exact. What
about similar modeling studies in other regions, e.g., the southern part of China,
Northern America, and Europe? For example, Sarwar et al. (2012, 2014) developed the
same model, CMAQ, to evaluate the effect of $ClNO_2$ production on air quality,
including the total nitrate, in the US and the Northern Hemisphere. However, these two
critical papers are not discussed anywhere in the current manuscript.

**Response #10:** Thanks for your comments. As this is not a review paper, our intention
is to include the most relevant studies in this region. We included some discussion of
Sarwar et al. (2012, 2014) as requested on page40, lines 537-543.

**Comment #11:** Line 73, This statement might be true, but the authors did not provide
any evidence/reference to support it.

**Response #11:** This sentence is removed.

**Comment #12:** Line 77-80. This statement is not correct. For example, Wang et al.
(2016) and Brown et al. (2016) reported extremely high $N_2O_5$ mixing ratios at a site in
Hong Kong (a coastal city) of up to 8ppbv (1min average) or 12ppbv (1min average).
This brings up another issue. Should include the average time when report observational
results, e.g., 1 s average, 1 min average, or 1h average.

**Response #12:** High concentrations of N2O5 in Hong Kong is likely affected by non-
local emissions from city clusters in the Pearl River Delta (PRD) region during some
high pollution episodes. We remove the relative clause, "which were significantly
higher than those in unpolluted coastal cities and the lower atmosphere in the remote
Arctic region." in the revised manuscript. We agree with the reviewer that it is necessary
to point out the averaging time when describing the concentrations and they are
included in the revised manuscript.

The sentences in Line 77-80 are revised as "According to the field measurements in
June 2017 in Beijing (Zhou et al., 2018), the 2-min averaged concentrations of reactive
$Cl_2$ and $ClNO_2$ reached up to 1000 pptv and 1200 pptv, respectively, during some severe air pollution period in summer. The corresponding concentrations of $N_2O_5$ and nitrate reached as high as 700 pptv (2 min average) and 5 μg m$^{-3}$ (5 min average) from about 40 pptv and 1 μg m$^{-3}$"

**Comment #13:** Line 79-80. There is no Li et al. (2017) in the reference list. Are you referring to Li et al. (2016)? That is not a proper reference here, because that paper is a modeling study that used the measurement results from Wang et al. (2016).

**Response #13:** Sorry for my carelessness. As the reviewer described, Li et al. (2017) should be Li et al.(2016). We revise it throughout the whole manuscript and here we remove it.

**Comment #14:** Line 82. These references are not the right ones here. The first measurements of ClNO2 in the real atmosphere, Osthoff et al. (2008) and Thornton et al. (2010), are better ones.

**Response #14:** Revised.

**Comment #15:** Line 102. This is not entirely true. For instance, Hossaini et al. (2016) developed a global chemical transport model (TOMCAT) and included several heterogeneous reactions of chlorine species on chloride-containing aerosol.

**Response #15:** The reviewer might misread the sentence. We did include the fact that some models have some heterogeneous reactions by saying that "generally missing" and "in most" CTMs. No changes were made regarding this comment.

**Comment #16:** Line 107-111. I assume the replacement is only in Beijing city but not in the surrounding areas. Is that correct?

**Response #16:** Yes, replacing coal with natural gas only occurred in Beijing. Reduction of coal consumption in surrounding regions was less than 15% for most other provinces and cities and there were no strict control measures for biomass burning (except Hebei), cooking and municipal solid waste incineration yet. Thus, the Cl emissions estimated for 2014 by Fu et al. (2018) were used for other areas. This is explained in the revised manuscript on page 30, lines 223-228.

**Comment #17:** Line 117-118. This sentence is a bit out-of-blue. The following sentence makes more sense to me.
'Thus, it is important to compile an updated emission inventory for Beijing (and its surrounding areas) to include the emissions from cooking and other sources (coal burning, solid waste burning, biomass burning, etc.)."

**Response #17:** Thanks for your constructive comment. This sentence is revised to read "Thus, it is necessary to compile an updated emission inventory for Beijing to include the emissions from cooking and other sources (coal burning, solid waste burning, biomass burning, etc.) in order to explore the chlorine species emission on atmospheric nitrate formation."

**Comment #18:** Line 128-130. Should add some reference here. Also, 'NH3' should also be considered as a common species.

**Response #18:** This seems to be a common knowledge among air quality modelers, but we included a citation (Wang et al., 2014) per reviewer's request. $NH_3$ is added to the sentence and its emission is also summarized in the revised manuscript in Table S1.

**Comment #19:** Line 136. Should mention the number (from 2000 Mt in 2014 to 490 Mt in 2017) here.

**Response #19:** This is now included in the revised manuscript.

**Comment #20:** Line 142. Should list the emission factors for different sectors, at least in the supplement. Also, give the reference.

**Response #20:** There are quite a number of different emission factors used in the calculation, which have already been summarized in Table 3 of Fu et al. (2018). We added the citation in the revised manuscript.

**Comment #21:** Line 156-157. Should provide reasons why you chose this number of hours. Three hours of cooking time seem to be a bit long for me. Also, 'restaurant' should be 'social cooking', is that correct?

**Response #21:** Sorry, it's a typo. It should be 0.5 h following the study by Wu et al (2018). based on a survey data. The emissions were correctly calculated using 0.5 h. Also, 'restaurant cooking' has been revised to 'commercial cooking'.

**Comment #22:** Line 160. Any reason that you chose '150'?

**Response #22:** It's based on Wu et al. (2018). Citation is now included.

**Comment #23:** Line 173-174. A brief description is needed for other emissions, which are the precursors of N2O5, ClNO2, and nitrate aerosol.

**Response #23:** We supplement the description of emission in page 30, line 229-231 and this sentence is revised as "Emissions of conventional species for this study period are developed in a separate study that is currently under review and are summarized in Table S1."

**Comment #24:** Section 2.2. The authors added several reactions to the CMAQ model, and this seems to be one of the major contributions of this study. **[this comment seems to be less coherent, so we break it into several sentences and address them individually]**

(1) However, what is the difference between the mechanism in the current study and that in Sarwar et al. (2012, 2014)?

(2) I notice that one of the co-authors in the present study is also a co-author of Sarwar et al. (2014).

(3) I strongly advise the authors to carefully review the previous works and identify the advantage of the current work, instead of avoiding the comparison between the current study with the previous ones.

(4) Besides, did you compare your scheme with Zheng et al. (2015)?

**Response #24:**

**(1)** Sarwar et al. (2012, 2014) only consider the reaction of $N_2O_5$ with PCl. They did not include those heterogeneous reactions involving $Cl_2$ production (the reaction of $O_3$, OH, HOCl, $ClNO_2$ and $ClONO_2$ with PCl).

**(2)** It is correct that one of the authors of Sarwar et al. (2014) happens to be a co-author of this study. We didn't understand why this comment is even relevant, so no changes were made regarding this comment.

**(3)** Previous works were reviewed in the introduction section and discussed throughout the manuscript wherever appropriate.

**(4)** The major difference in Zheng et al.'s treatment of heterogeneous chemistry and our approach is that they chose to use an empirical expression for RH dependent uptake coefficients of NO2 and SO2. There is no evidence so far that that RH-dependent expression is any better than simple constant values. Thus, it is out of the scope of this paper to compare Zheng et al.

**Comment #25:** Line 178. What do you mean by 'current CMAQ model'? Is 'Zheng et al. (2015)' a proper reference for 'current CMAQ model'?

**Response #25:** Zheng et al.(2015) was not the right reference. The current CMAQ model refers to the one used by Hu et al. (2016) and Hu et al. (2017)

Hu, J., Chen, J., Ying, Q., Zhang, H., 2016. One-Year Simulation of Ozone and Particulate Matter in China Using WRF/CMAQ Modeling System. Atmos. Chem. Phys. 16, 10333-10350.

Hu, J., Wang, P., Ying, Q., Zhang, H., Chen, J., Ge, X., Li, X., Jiang, J., Wang, S., Zhang, J., Zhao, Y., Zhang, Y., 2017. Modeling biogenic and anthropogenic secondary organic aerosol in China. Atmos. Chem. Phys. 17, 77-92.

**Comment #26:** Line 182. Li et al. (2016) is not a proper reference for this equation. Should refer to Bertram and Thornton (2009) or Roberts et al. (2009) with a very similar formula. This brings up another persistent issue that you should use 'Bertram and Thornton (2009)' instead of 'Bertram et al. (2009)'. There are only two authors to that paper. Please check the manuscript for this error.

**Response #26:** The reference 'Li et al. (2016)' is replaced with 'Bertram and Thornton (2009)' and we revise the 'Bertram et al. (2009)' with 'Bertram and Thornton (2009)' throughout the manuscript.

**Comment #27:** Line 177. H2O means water vapor. Is that right?

**Response #27:** Yes.

**Comment #28:** Line 220-22. This part is a bit confusing. Why did you call the values 'preliminary'? Did you get these data from the reference (Keene et al., 1999)? Or did you make a guess on these values? I see that you made some sensitivity cases later. Perhaps you should mention that here.

**Response #28:** In Keene et al. (1990) [It was incorrectly cited in the original manuscript as Keene et al. (1999). We apologize for this mistake.], the daytime uptake coefficient of $O_3$ was not based on direct experimental measurements but was estimated indirectly based on a steady-state analysis of $Cl_2$ production rate in a hypothesized geochemical cycle of reactive inorganic chlorine in the marine boundary layer. The estimated daytime O3 uptake coefficient was around $10^{-4}$ to $10^{-3}$. Lower nighttime value was further estimated based on the observation of lower $Cl_2$ production in the marine boundary layer at night. In this study, we used the values used by Keene et al. (1990) in their simulations. As both daytime and nighttime values may have significant uncertainties, we choose to call the values used in this study "preliminary". We add the sentence 'it's a simulation-based result, which presents high uncertainty' behind the above sentence.

Due to the high uncertainty of $O_3$ uptake coefficient, we do some sensitivity case to evaluate how the effect of this uncertainty on the underestimation of atmospheric $Cl_2$ concentration. This is now mentioned right after the sentence in question.

**Comment #29:** Line 229. Just out of curiosity, is the gas-phase chemistry of chlorine the same as that in Sarwar et al. (2012)?

**Response #29:** The gas-phase chemistry used by Sarwar (2012) is not as complete. It only has 9 inorganic reactions while the one we used in our study includes 22 inorganic reactions. Most reactions of ClONO and ClONO2 are missing from Sarwar (2012). In addition, Sarwar et al. (2012) used CB05 but we used SAPRC11. The organic reactions are also different but to less a degree.

**Comment #30:** Section 3.1. What about the model performance on the simulation of NO2, O3, and PM2.5? These are particularly important to the formation of nitrate aerosol, the sole focus of the current work.

**Response #30:** Model performance of these species were included in the revised manuscript. See our response to comment 3 for more details.

**Comment #31:** Line 251-252. This treatment artificially amplifies the heterogeneous reaction rate by 5 and 10 times. Is it logical to do that based on the comparison with one measurement data set? How does the CMAQ model perform in general in the simulation of surface area? Is it a common issue? Or did it only happen in your simulation? How is the simulation of $PM_{2.5}$? This highlights the need for the evaluation of the routine air pollutants, e.g., $PM_{2.5}$.

**Response #31:** Figure R1a below is the detailed comparison of our modeled surface area with observations averaged between June 11-15, 2017. Under-prediction of surface area varies between 6-12 times. CMAQ model performance for surface area has not been extensively studied. In the only study we found in the literature (Park et al., 2006), similar under-prediction of particle surface area was reported. In their study, the mass concentration of PM2.5 at Atlanta, United State is generally well reproduced by the CMAQ model but the surface area shows large under-predictions similar to what we have seen in our study (see Figure R1b).

[Figure]

Figure R1a (left panel): Predicted averaged hourly particle wet surface area at IAP (June 11-15, 2017) and the ratio of observed to predicted PSA (O/P). R1b (right panel): CMAQ predicted and observed daily particle surface area, volume, and mass concentrations for PM2.5 from 1/1/1999 to 8/31/2000 at the JST station in Atlanta, Georgia. (R1b is adapted from Park et al., 2006).

**Reference:**
Park, S.K., Marmur, A., Kim, S.B., et al.: Evaluation of fine particle number concentrations in CMAQ. Aerosol Science and Technology, 40, 985-996, 2006.

**Comment #32:** Line 255. $ClNO_2$ is underestimated in your cases. Does it mean that nitrate aerosol is over-predicted?

**Response #32:** Nitrate aerosol is also slightly underpredicted most of the hours. It was
over-predicted slightly on the night on June 13. Nitrate concentrations are affected by
many other factors so a simple anti-correlation between ClNO2 and nitrate cannot be
assumed. No changes were made regarding this comment.

**Comment #33:** Line 258. The $O_3$ will increase or decrease if you change the uptake
coefficient of $O_3$. This also calls for the model evaluation on $O_3$.

**Response #33:** The impacts of heterogeneous chlorine chemistry on O3 formation are
complicated. On one hand, $O_3$ is consumed by the heterogeneous reaction with PCl. On
the other hand, the generated Cl2 photolyze to produce Cl atom, resulting in the increase
of $O_3$. The impact of chlorine chemistry on ozone is a very important by itself and has
been explored in a sperate manuscript that is currently under review. Model
performance of $O_3$ is evaluated by comparing with observations at 12 sites in Beijing
(The average NMB/NME values for $O_3$ across the 12 sites are -8%/29%)

**Comment #34:** Line 260. Several studies have been reported that the parameterization
used in the present study (Bertram and Thornton, 2009) has some uncertainty, including
Tham et al. (2018), McDuffie et al. (2018a), McDuffie et al. (2018b) and the references
therein. How these uncertainties affect the conclusions should be discussed.

**Response #34:** We investigated the uncertainty in the predicted nitrate concentrations
using the parameterized $N_2O_5$ uptake coefficients of Bertram and Thornton (2009) by
using two sensitivity simulations in the original manuscript. In one simulation, the
parameterization of Davis et al. (2008), which is the default $N_2O_5$ parametrization
scheme in CMAQ 5.0.1, was used. It generally yields slightly lower $\gamma_{N_2O_5}$ than the

Bertram and Thornton (2009). In the other simulation, the $\gamma_{N_2O_5}$ was fixed at a constant value of 0.09, which is the maximum value derived by Zhou et al. (2018) based
on summertime field measurement in urban Beijing. On average, it is 4-6 times higher
than those based on Bertram and Thornton (2009). Table 3 in the original manuscript summarized the $\gamma_{N_2O_5}$ averaged for each day and night from these simulations and the corresponding nitrate concentrations. Predicted nitrate concentrations are sensitive to changes in the changes in $\gamma_{N_2O_5}$, with approximately 50% increase in the nitrate when the $\gamma_{N_2O_5}$ is fixed at 0.09. The discussion of these two sensitivity simulations are slightly revised from the original manuscript and can be fond in the revised paper on
page 37, lines 431-434.

**Comment #35:** Line 277-278. The underestimated nitrate concentration could be due to many reasons. For example, is NO2 underestimated as well? This is another reason to show the evaluation of NO2 simulation. The uncertainty of the parameterizations of N2O5 uptake and ClNO2 yield (comment 31) could also be applied here. Besides, how did you simulate the most critical OH precursor, i.e., HONO? Did you evaluate your HONO simulation? Did you consider the NO2 uptake on environmental surfaces? What about other HONO sources? Several recent papers have shown that HONO is very important in simulating nitrate, e.g., Fu et al. (2018).

**Response #35:** We have evaluated the model performance of $NO_2$, which shows that the $NO_2$ concentration isn't significantly underestimated (The average NMB/NME values for $NO_2$ across the 12 sites are -7%/59%). In original CMAQ, the $NO_2$ hydrolysis produces HONO and $HNO_3$. However, in the improved CMAQ, this reaction is revised as:

$2NO_2(g) + Cl^-(aq) \rightarrow ClNO(g) + NO_3^-(aq)$ (if the NO2 is redundant, $2NO_2(g) + H_2O(aq) \rightarrow HONO(g) + NO_3^-(aq)$ ).

As for HONO (assuming the HONO is produced in nighttime), the CMAQ model have covered the reaction of HONO photolysis to produce OH. However, HONO photolysis affects the OH level just a few hours in the morning and can be neglected.

**Comment #36:** Line 281-282. What do you mean by 'the nitrate level is higher in the daytime and lower in nighttime'?

**Response #36:** It's redundant. We have deleted it.

**Comment #37:** Line 290-291. It may be true that the uptake coefficients for these two molecules are the most important parameters. But what about the uptake coefficients for other species/process. Did you run any sensitivity cases to examine that?

**Response #37:** We haven't run any sensitivity cases to examine the impact of the uptake coefficient of other species on nitrate. As we demonstrated in manuscript, the gas-to-particle partitioning of $HNO_3$ and the reaction $N_2O_5$ with PCl are the major pathways of producing nitrate in daytime and nighttime, so we choose to run sensitivity cases of $O_3$ ($O_3$ uptake is major contributor to $Cl_2$ in R13-R17) and $N_2O_5$ uptake coefficients.

**Comment #38:** Line 293-295. Or maybe just because the direct emission of Cl2 is not well represented in the emission inventory

**Response #38:** The underprediction is *unlikely* due to missing primary $Cl_2$ emissions. It is generally accepted that direct $Cl_2$ emissions from power plants or residential coal burning are in a smaller quantity (less than 3% in total Cl, Deng et al., 2014). In addition, Liu et al. (2017) revealed that there is only a weak correlation between $Cl_2$ with other primary emission indicators ($K^+$ for biomass burning (R=0.004), $SO_2$ for power plant emissions (R=0.31) and NOx representing transportation emissions (R=0.01)) or
precursors (HCl (R=0.08) and PCl (R=0.01))

Deng, S., Zhang, C., Liu, Y., et al.: A Full-Scale Field Study on Chlorine Emission of Pulverized
Coal-Fired Power Plants in China. Research of Environmental Science. In Chinese, 27, 127-133,
2014.

**Comment #39:** Line 316. $N_2O_5$ uptake process is very complicated. The word,
'inferior', is definitely not the one I would use to describe the parameterization based
on nitrate and sulfate. concentration. Please revise. The uncertainty of the
parameterizations of $N_2O_5$ uptake and $ClNO_2$ yield also works here.

**Response #39:** We completely agree with the reviewer.

**Comment #40:** Section 3.3. I advise to include the simulated results of NO2, O3, and
PM2.5.

**Response #40:** We have included them in SI Figure S3, which also be shown in Figure
R2.

[Figure]

[Figure]

[Figure]

**Figure R2 The spatial distribution of NO₂(a), O₃(b) and PM₂.₅(c) concentration averaged in 11 to 15, June.**

**Comment #41:** Line 328. Should include a figure for the 'intensive emissions of chlorine species'

**Response #41:** We have included it in Figure S2, which is also represented in Figure R3.

[Figure]

**Figure R3 The spatial distribution of PCl emission in Beijing in 2017 (Unit: Kg/year per grid).**

**Comment #42:** Line 328-330. The presence of a high concentration of ClNO2 and Cl2 away from the fresh emissions does not necessarily mean that ClNO2 and Cl2 are easy to transport.

For example, the production of ClNO2 requires the presence of chloride, NO2, and O3. In the areas close to the fresh emissions, O3 is commonly low, and the production of NO3 (hence N2O5 and ClNO2) is limited. Therefore, the production of ClNO2 is generally not found near fresh emissions.

As to the Cl2, perhaps the contribution of direct emission to the level of Cl2 is not significant, and Cl2 is predominantly produced in the atmosphere. So the high levels of Cl2 are found away from the fresh emissions.

**Response #42:** We agree with the reviewer's opinion and include this comment in page 37, line 444-450 in manuscript.

**Comment #43:** Line 336. Why 'more N2O5 is converted into nitrate'? Are you implying that the uptake coefficient calculated with the Bertram and Thornton (2009) is higher than that with Davis et al. (2008)?

**Response #43:** Yes, this is indeed the case. Table 3 compares the uptake coefficient of N2O5 based on the two parameterization and clearly shows that the Bertram and Thornton equation generally gives higher uptake coefficients. This is also consistent with the conclusion of McDuffie et al. (2018). We add this explanation in revised manuscript on page 38, lines 456-460.

**Comment #44:** Line 351-352. Was the observation in PKU conducted in the same period?

**Response #44:** No, the observation in PKU is conducted in November. This is clarified in the revised manuscript.

**Comment #45:** Line 355-357. Are you implying that in cleaner days, the OH level is higher, so the production of HNO3 from OH + NO2 is more important?

**Response #45:** The sentence in question discusses the *relative* importance of the homogeneous and heterogeneous pathways in nitrate formation. The difference is likely due to a combination of   higher OH concentrations in this study and more surface areas available for heterogeneous reaction in the winter during the PKU study. The difference in OH level between the two studies (this study vs. the PKU study) is mainly driven by the seasonal variation of the solar radiation. This is clarified in the revised manuscript.

**Comment #46:** Line 371-372. How did you treat the reaction of NO2 + H2O (aq)? Did you revise it to NO2+ Cl- or did you use both?

**Response #46:** Both reactions are included. The NO2 + Cl reaction is only considered when Cl concentration is greater than zero. No changes were made regarding this comment.

**Comment #47:** Line 396-397. It is correct that the emission of chlorine species is vital to chlorine chemistry study. But the current study does not demonstrate this point.

**Response #47:** We agree with the reviewer on this. This sentence is removed in the revised manuscript.

**Comment #48:** Line 402-404. Not necessarily correct. See comment above.

**Response #48:** The sentence 'Cl$_2$ and ClNO$_2$ are easy to transport among cities because high concentrations of them are not found in southern region with intensive emissions of chlorine species.' is revised as 'High concentration of Cl$_2$ and ClNO$_2$ are not found in southern region with intensive emissions of chlorine species may be related to high O3 concentration generally occurred in suburban'

**Comment #49:** Line 409-411. What implications? Care to elaborate? See comment 2 for example. But more thoughts are definitely of value to the policymakers.

**Response #49:** The sentence 'This study aims to improve our understandings on the chlorine chemistry and its impact on nitrate formation, which can provide useful implications on the nitrate pollution control strategies for those regions that suffered serious nitrate pollution.' is revised as 'This study aims to improve our understandings on the chlorine chemistry and its impact on nitrate formation, The chloride chemical mechanism study in this work indicates that not only the $NO_X$ emission is needed to be controlled, but also the emission of reactive chlorine species should be limited as well in order to alleviate the nitrate pollution'

**Comment #50:** The reference list is not organized according to the alphabet. For example, L is before K, J is after K, Rudich is before Roberts, and Spicer is before Song.

**Response #50:** Revised

**Comment #51:** Table 2. What are the effects of R6, R11, R13-R18 on the production of nitrate aerosol? Also, please provide the reference for all reactions.

**Response #51:** The reactions R6 and R11 directly affect the nitrate and R13-18 indirectly affect it by elevating the OH level due to production of $Cl_2$. This discussion is included in the revised manuscript on page 27, line 108-111. The references for all reaction have included in Table 3.

**Comment #52:** Line 38. 'photolyze' is a better word than 'decompose' since it is a photolysis reaction.

**Response #52:** The ClNO2 reacts with particle surface to form nitrate, which is not a photolysis reaction. It is changed to 'reacts with particle surface' to make it more specific.

**Comment #53:** Line 62. Should add '(aq)' after 'H2O'

**Response #53:** Revised

**Comment #54:** Line 63. should define CMAQ and WRF-Chem here.

**Response #54:** Revised

**Comment #55:** Line 81. should be 'chloride-containing', not 'chlorine-containing'.

**Response #55:** Revised

**Comment #56:** Line 119. The definition of CMAQ should be moved to line 63.

**Response #56:** Revised

**Comment #57:** Line 130. Add a space between 'emissions' and 'were'.

**Response #57:** Revised

**Comment #58:** Line 145. 'EF represents the emission factor' should be 'EFi,j represents the emission factor of pollutant j in sector i'.

**Response #58:** Revised

**Comment #59:** Line 148-149. 'had been detailed described' should be 'had been described in detail'

**Response #59:** Revised

**Comment #60:** Line 158. Add 'from social cooking' after 'Vc is the volume of exhaust gas'.

**Response #60:** Revised

**Comment #61:** Line 161. 'chose' should be 'chosen'

**Response #61:** Revised

**Comment #62:** Line 166. Delete 'that'

**Response #62:** The sentence has been deleted

**Comment #63:** Line 168. Use the same decimal for all data.

**Response #63:** Revised

**Comment #64:** Line 169. 'Others' should be 'Other'.

**Response #64:** Revised

**Comment #65:** Line 169. Add the publication year after 'Fu et al.'

**Response #65:** Revised

**Comment #66:** Line 169 and line 171. Two 'finally'?

**Response #66:** The redundant 'finally' has been deleted.

**Comment #67:** Line 185. 'Laboratorial' should be 'laboratory'

**Response #67:** Revised

**Comment #68:** Line 192. Do you mean 'equation (5)', instead of 'equations (2)'?

**Response #68:** It should be equation (5).

**Comment #69:** Line 198. Do you mean 'equation (6)', instead of 'equation (3)'?

**Response #69:** It should be equation (6).

**Comment #70:** Why did you use different terms for velocity in equation 5 and equation 8?

**Response #70:** The equation 5 has been revised as '$v$'.

**Comment #71:** Line 211. How did you calculate Kh, Kf, K3/K2, and K4/K2? Are they constants? If so, please add the number.

**Response #71:** These parameters have been demonstrated. As $K_h$ represents the dimensionless Henry's law coefficient ($K_h = [N_2O_5]_{aq}/[N_2O_5]_g = 10e(30)$). $K_f$ represents a parameterized function based on water concentration ( $K_f = 1.15e^6(1 - e^{-1.3e^{-1}[H_2O(l)]})$ and $K_3/K_2$ and $K_4/K_2$ are constants obtained by fitting data, which are $6 \times 10^{-2}$ and 29.

**Comment #72:** Line 216 and line 208. Use the consistent form for units. m3/m3 or m3 m-3.

**Response #72:** Revised

**Comment #73:** Line 227. Which year?

**Response #73:** This sentence is revised as 'These heterogeneous reactions of chlorine are incorporated into revised CMAQ (version 5.0.1) to simulate the distribution of nitrate concentration in Beijing from 11 to 15 June 2017'

**Comment #74:** Line 232. It is weird to see 'Figure 3' before 'Figure 1 and 2'. Maybe a map with three domains in the supplement as Figure S1 is better.

**Response #74:** A map with three domains is included as Figure S1(Figure R4 below):

[Figure]

**Figure R4 the three nested domain setting in this work.**
**Comment #75:** Line 232. '40ºE' should be '40ºN'.
**Response #75:** Revised
**Comment #76:** Line 296. Remove the extra space between 'empirical' and 'and'.
**Response #76:** Revised
**Comment #77:** Line 315. Add the year for 'Davis et al.'. Check through the manuscript
for a similar issue.
**Response #77:** This sentence has been deleted in the revised manuscript.
**Comment #78:** Line 350. 'are produced' should be 'is produced'.
**Response #78:** Revised
**Comment #79:** Line 369. Add 'of' between 'uptake' and 'N2O5'.
**Response #79:** Revised
**Comment #80:** Line 378. 'even through' should be 'even though'.
**Response #80:** Revised
**Comment #81:** Line 396. 'This' should be 'These'.
**Response #81:** Revised

**Comment #82:** Line 397. 'becase' should be 'because'.

**Response #82:** Revised

**Comment #83:** Line 397. 'the cornerstones' should be 'the cornerstone' or 'one of the cornerstones'.

**Response #83:** Revised

**Comment #84:** Line 401. 'chlorine' should be 'chloride'.

**Response #84:** Revised

**Comment #85:** Line 409. 'understnadings' should be 'understandings'.

**Response #85:** Revised

**Comment #86:** Figure 3. Should point out the area of BJ and the location of the sampling site. In d, f, and h, should use $\Delta N_2O_5$, $\Delta NO_3-$, and $\Delta NO_3-$, instead of $N_2O_5$, $NO_3-$, and $NO_3$

**Response #86:** The area of BJ and the location of the sampling site is labeled in Figure S1. We have used $\Delta N_2O_5$, $\Delta NO_3-$, and $\Delta NO_3-$, instead of $N_2O_5$, $NO_3-$, and $NO_3$.

**Comment #87:** Figure 4. In the sub-plot Daytime Gas-phase, the title of Y-axis should be '$HNO_3$ production rate (ppt h-1)'. The same revision should be applied to the sub-plot Nighttime Gas-phase. The sub-plot Nighttime Heterogeneous, the title of Y-axis should be 'Nitrate production rate ($\mu g$ m-3 h-1)'. No sub-plot daytime Heterogeneous?

**Response #87:** Revised. Additionally, the heterogeneous reaction of $NO_2$ with PCl have less contribution to diurnal nitrate (less than 2%) because the extremely lower uptake coefficient. In addition, we supplement the daytime heterogeneous.

**Reviewer #2**

**Comment #1:** The manuscript of Qiu et al., reported the influence of chloride chemistry
to particulate nitrate formation in the framework of CMAQ. The authors included
several chloride reactions into the CMAQ chemical mechanism. This topic is in the
scope of ACP, and would benefit the knowledge of the heterogeneous reactions in the
formation of particulate nitrate in Beijing, China. While in this paper, the detail of the
parameters lacking the foundation and the analysis of model result seems common.
Some of the conclusions seems can't obtain from the model result analysis (see major
comments). The following comments should be addressed before publishing in ACP.

**Response #1:** Thanks for the positive comments on this manuscript. The suggestions
are addressed in detail in the following.

**Comment #2:** Line 250-252, the treatment of aerosol surface area by time 5 or 10 in
this model is unconvinced. Should provide more evidence to support the rationality.

**Response #2:** Figure R1a below is the detailed comparison of our modeled surface area
with observations averaged between June 11-15, 2017. Under-prediction of surface area
varies between 6-12 times. CMAQ model performance for surface area has not been
extensively studied. In the only study we found in the literature (Park et al., 2006),
similar under-prediction of particle surface area was reported. In their study, the mass
concentration of PM2.5 at Atlanta, United State is generally well reproduced by the
CMAQ model but the surface area shows large under-predictions similar to what we
have seen in our study (see Figure R1b).

[Figure]

Figure R1a (left panel): Predicted averaged hourly particle wet surface area at IAP (June
11-15, 2017) and the ratio of observed to predicted PSA (O/P). R1b (right panel): CMAO
predicted and observed daily particle surface area, volume, and mass concentrations for PM2.5 from

1/1/1999 to 8/31/2000 at the JST station in Atlanta, Georgia. (R1b is adapted from Park et al., 2006).

**Comment #8:** Line 289 the section title should not be the estimation of uptake
coefficients of O3 and N2O5, but the influence of the change of these parameters.

**Response #8:** The title is revised as 'Impact of uptake coefficients of $O_3$ and $N_2O_5$ on
chlorine species and nitrate'

[revised manuscript text omitted]

The rate of $ClNO_2$, the change of $ClNO_2$

includes both removal and production terms, as shown in equation (5):

$$\frac{d[\text{ClNO}_2]}{dt} = -k_i^{\text{I}}[\text{ClNO}_2] + k_6^{\text{I}}\phi_{\text{ClNO2}}[\text{N}_2\text{O}_5] \tag{$6$5}$$

$$= -k_i^{\text{I}}[\text{ClNO}_2] + k_6^{\text{I}}\phi_{\text{ClNO2}}[\text{N}_2\text{O}_5]_{t0}\exp(-k_6^{\text{I}}t)$$

Assuming $\phi_{\text{ClNO2}}$ is a constant, analytical solution  for equation (5) can be obtained, as shown in equation (6):

$$$$
$$$$
$$= [\text{ClNO}_2]_{t0}\exp(-k_i^{\text{I}}\Delta t)$$
$$+ \frac{k_6^{\text{I}}\phi_{\text{ClNO2}}[\text{N}_2\text{O}_5]_{t0}}{k_i^{\text{I}} - k_6^{\text{I}}}[\exp(-k_6^{\text{I}}\Delta t) - \exp(-k_i^{\text{I}}\Delta t)] \tag{$7$6}$$

where $k_i^{\text{I}}$ represents the pseudo first-order rate coefficient of either reaction R17 or R18, depending on pH.

The uptake coefficients $\gamma$ of gaseous species are obtained from published laboratorial studies. In the original CMAQ, the uptake coefficient of $\text{N}_2\text{O}_5$ is determined as a function of the concentrations of $(\text{NH}_4)_2\text{SO}_4$, $\text{NH}_4\text{HSO}_4$ and $\text{NH}_4\text{NO}_3$ (Davis et al.,

2008). In this study, the PCl and $\text{NO}_3^-$ dependent parameterization by Bertram and Thornton (2009) (equation (7)) is used

:

$$ \begin{cases}  \\  \end{cases} \tag{$8$7}$$

$$= \begin{cases} 0.02, \quad for\ frozen\ aerosols \\ 3.2 \times 10^{-8}K_f\left[1 - \left(1 + \dfrac{6 \times 10^{-2}[\text{H}_2\text{O}]}{[\text{NO}_3^-]} + \dfrac{29[\text{Cl}^-]}{[\text{NO}_3^-]}\right)^{-1}\right] \end{cases}$$

In the above equation,

$K_f$

is parameterized function based on underline molarity of water: $K_f =$

[revised manuscript text omitted]

The  uptake coefficient of $O_3$  could  be an important factor affecting the predicted $Cl_2$ concentrations as it is found that the heterogeneous reaction of $O_3$ is the major source of $Cl_2$ during this period  (see discussion in Section 3.2).  The influence of different parametrizations of the uptake coefficient of $N_2O_5$ on $ClNO_2$ and nitrate concentrations are also discussed in Section 3.2~~). The improved CMAQ can accurately capture the diurnal variation of $N_2O_5$ concentration as well as the peak values (Figure 1(c)). In general, although the overall NMB and NME of BASE case (-20% and 38%) are slightly better than the HET case (-21% and 41%), the improved CMAQ  (with the NMB and NME of -3% and 14%) perform better than original CMAQ (with the NMB and NME of -33% and 52%) in some period of heavy air pollution (such as the nighttime on 12 June and 13 June).~~.

[revised manuscript text omitted]
 ~~are important becase it is the cornerstones of studying chlorine chemistry; (2) The sensitivity analysis shows that a non-constant parameterization of the uptake coefficients of O₃ that consider the influence of PCl concentrations, meteorology conditions, etc., might be needed, N₂O₅ uptake coefficient expressed as a function of the concentrations of chlorine can capture the nitrate concentration better than others; (3) Cl₂ and ClNO₂ are easy to transport among cities because high concentrations of them are not found in sourthern. (4) more importantly, currentsignificantlythe nitrate levelthe reaction rate ofwith NO₂formation~~pathways due to missing chlorine heterogeneous chemistry.

***Data availability***. The data in this study are available from the authors upon request (shxwang@tsinghua.edu.cn)

*Author contributions*. XQ, QY, SW, and JH designed the study; YS, BL, AS, XY provided observation data; XQ, QY, SW, JZ, QX, DD, LD and JX analyzed data. XQ, QY and SW wrote the paper.

*Competing interests*. The authors declare that they have no conflict of interest.

*Acknowledgments.* This work was supported by National Natural Science Foundation of China (21625701), China Postdoctoral Science Foundation (2018M641385), National Research Program for Key Issue in Air Pollution Control (DQGG0301, DQGG0501) and National Key R&D Program of China (2018YFC0213805, 2018YFC0214006). The simulations were completed on the "Explorer 100" cluster system of Tsinghua National Laboratory for Information Science and Technology.

[revised manuscript text omitted]

), (f) daytime
nitrate and (g) nighttime nitrate. Units are µg m$^{-3}$.

[Figure]

Figure 4. Contributions of different  homogeneous and heterogeneous pathways to nitrate formation.

[Figure]

[Figure]

Figure 1

[Figure]

Figure 2

[Figure]

Figure 3
Note: the distribution of Cl₂ and ClNO₂ in HET minus BASE have not been shown because their concentrations in BASE case are rather low (close to 0)

[Figure]

Figure 4

Table 1 The sectoral emissions of HCl, Cl$_2$ and PCl in Beijing in 2017. Unit: Mg year$^{-}$
$^{1}$

| Sector | Emissions | | |
|---|---|---|---|
| | HCl | Cl$_2$ | PCl |
| Power plant | 22.8 | 1.2 | 6.75 |
| Industry | 587.3 | 20.1 | 89.2 |
| Residential | 202.4 | 8.1 | 34.7 |
| Biomass burning | 0.182 | 0 | 0.14 |
| Municipal solid waste | 1080.2 | 0 | 8.47 |
| Cooking | 0 | 0 | 426.8 |
| Total | 1892.9 | 29.4 | 566.1 |

Table 2 Major gas-phase and heterogeneous pathway of producing nitrate in original CMAQ and newly added or revised heterogeneous reactions in improved CMAQ.

| Type | Reactions | No. | Reference | Comment |
|---|---|---|---|---|
| **Original CMAQ** | | | | |
| Gas-phase chemistry | $OH + NO_2 \rightarrow HNO_3$ | R1 | | |
| | $N_2O_5 + H_2O \rightarrow$ $2HNO_3$ | R7 | | |
| | $HO_2 + NO_3 \rightarrow$ $0.2HNO_3 + 0.8OH\cdot + 0.8NO_2$ | R8 | | |
| | $NO_3 + VOCs^a \rightarrow HNO_3$ | R9 | | |
| Heterogeneous chemistry | $N_2O_5\text{(g)}$  $+ H_2O\text{(aq)} \rightarrow$ $2H^+ + 2NO_3^-$ | R5 | | |
| | $2NO_2\text{(g)} + H_2O\text{(aq)} \rightarrow$ $HONO\text{(g)} +$ $H^+ + NO_3^-$ | R10 | | |
| **Improved CMAQ** | | | | |
| Newly added or revised heterogeneous reactions | $N_2O_5\text{(g)} + H_2O\text{(aq)} + Cl^-\text{(aq)} \rightarrow ClNO_2\text{(g)} + NO_3^-$ | R6 | Bertram and Thornton (2009) | Revise R5 |
| | $2NO_2\text{(g)} + Cl^-$ $\rightarrow ClNO\text{(g)} + NO_3^-$ | R11 | Abbatt et al. (1998) | Revise R10 |
| | $NO_3\text{(g)} + 2Cl^-$ $\rightarrow Cl_2\text{(g)} + NO_3^-$ | R12 | Rudich et al. (1996) | Increase $NO_3^-$ |
| | $O_3\text{(g)} + 2Cl^- + H_2O\text{(aq)} \rightarrow Cl_2\text{(g)} +$ $O_2\text{(g)} + 2OH^-$ | R13 | Abbatt et al. (1998) | Affect OH |
| | $2OH\cdot\text{(g)} +$$2Cl^- \rightarrow Cl_2\text{(g)} + 2OH^-$ | R14 | George et al. (2010) | Affect OH |
| | $ClONO_2\text{(g)} + Cl^-$ $+$  $\rightarrow Cl_2\text{(g)} +$ $NO_3^-$ | R15 | Deiber et al. (2004) | Affect OH |
| | $HOCl\text{(g)} + Cl^-$ $+ H^+$ $\rightarrow Cl_2\text{(g)} + H_2O$ | R16 | Pratte et al. (2006) | Affect OH |
| | $ClNO_2\text{(g)} + Cl^-$ $+ H^+$ $\rightarrow Cl_2\text{(g)} + HONO$ $< 2.0)$ | R17 | Riedel et al. (2012) | Affect OH |
| | $ClNO_2\text{(g)} + H_2O\text{(aq)} \rightarrow Cl^- + NO_3^- + 2H^+ (pH \geq 2.0)$ | R18 | Rossi (2003) | Increase $NO_3^-$ |

a: presents different VOCs species. In the SAPRC--11 mechanism, the VOCs species include CCHO (Acetaldehyde), RCHO (Lumped C3+ Aldehydes), GLY (Glyoxal), MGLY (Methyl Glyoxal), PHEN (phenols), BALD (Aromatic aldehydes), MACR (Methacrolein), IPRD (Lumped isoprene product species).

Table 3 Observed day (D) and night (N) NO$_3^-$ concentrations (Obs.) and predicted uptake coefficient of N$_2$O$_5$ ($\gamma_{N2O5}$) and nitrate concentrations (Pred.) using the parameterizations of $\gamma_{N2O5}$ by Bertram and Thornton (2009) (Scenario 1), Davis et al. ., (2008) (Scenario 2) and the upper-limit value derived by Zhou et al.. (2018 (Scenario 3)

| | NO$_3^-$ | Scenario1 | | Scenario2 | | Scenario3 | |
|---|---|---|---|---|---|---|---|
| | Obs. | $\gamma_{N2O5}$ | Pred. | $\gamma_{N2O5}$ | Pred. | $\gamma_{N2O5}$ | Pred. |
| 06/11-D | 2.54 | 0.033 | 1.59 | 0.008 | 1.32 | 0.09 | 2.17 |
| 06/11-12-N | 2.42 | 0.043 | 1.67 | 0.037 | 1.37 | 0.09 | 2.12 |
| 06/12-D | 3.39 | 0.028 | 2.16 | 0.032 | 2.74 | 0.09 | 3.13 |
| 06/12-13-N | 4.24 | 0.021 | 4.02 | 0.022 | 4.05 | 0.09 | 6.04 |
| 06/13-D | 2.57 | 0.012 | 1.18 | 0.008 | 1.06 | 0.09 | 2.47 |
| 06/13-14-N | 4.10 | 0.022 | 4.45 | 0.022 | 4.45 | 0.09 | 7.13 |
| 06/14-D | 0.95 | 0.001 | 1.34 | 0.001 | 1.33 | 0.09 | 1.64 |
| 06/14-15-N | 2.75 | 0.013 | 1.00 | 0.007 | 0.96 | 0.09 | 2.33 |
| 06/15-D | 0.75 | 0.001 | 0.66 | 0.001 | 0.66 | 0.09 | 1.11 |

---

## Author Response (AR2)

**Response to reviewers' comments**

Thanks to the reviewers for giving us very useful comments to improve our manuscript entitled "Modeling the impact of heterogeneous reactions of chlorine on summertime nitrate formation in Beijing, China" **(acp-2018-1270)**. Detailed responses to reviewers' comment are list below:

**Report #1**

**Comment 1:** Qiu et al., developed a revised regional CMAQ model, by updating several chloride contained reactions, to study the impact of chloride-related heterogeneous chemistry on summer particulate nitrate formation. This work is meaningful and improved the understanding of regional nitrate pollution as well as chloride chemistry. Several comments should be addressed before publishing in ACP.

**Response #1**: Thanks for the positive comments on this manuscript. The suggestions are addressed in detail in the following.

**Comment 2:** Section 3.2, the uptake coefficient of $O_3$ on chloride-contain aerosol were increased by a factor of 10 and showed the importance of O3 heterogeneous reaction and the Cl2 budget. The increasing of $O_3$ uptake coefficient may also strongly affect the O3 lifetime, an intercomparison of modeled and observed O3 need to be conducted to verify the rationality and prove the significance.

**Response #2**: In previous response #3 from the reviewer #1, we have performed the comparison of predicted $O_3$ concentration in HET case with observation, as Table R1. We further compare the predicted $O_3$ concentration in uptake coefficient increased by a factor of 10 (represent as Sim* in Table R1) with observation and simulation in HET case. The results show that $O_3$ concentrations in 'Sim*' column are slight lower than the 'Sim' column, which may indicate that the $O_3$ consumption by heterogeneous reaction is generally larger than the production owing to the $Cl_2$ formation. Moreover, these differences are mainly found in nighttime because the weak photolysis limit the generated $Cl_2$ to transfer to $O_3$.

Table R1 Comparison of simulated 1-h $NO_2$, $PM_{2.5}$ and $O_3$ concentrations with observations averaged from 11 to 15, June, 2017 (Obs: observation, Sim:simulation).

| Sites | NO₂ | | | | O₃ | | | | | PM₂.₅ | | | |
|---|---|---|---|---|---|---|---|---|---|---|---|---|---|
| | Obs | Sim | NMB | NME | Obs | Sim | Sim* | NMB | NME | Obs | Sim | NMB | NME |
| WSXG | 49 | 54 | 11% | 55% | 99 | 122 | 121.6 | 23% | 63% | 40 | 38 | -6% | 53% |
| DL | 21 | 17 | -20% | 68% | 111 | 108 | 107.5 | -2% | 41% | 32 | 29 | -10% | 52% |
| DS | 47 | 53 | 13% | 54% | 100 | 114 | 113.8 | 15% | 56% | 44 | 41 | -7% | 53% |
| TT | 40 | 48 | 20% | 64% | 98 | 130 | 129.4 | 33% | 60% | 37 | 37 | 1% | 58% |
| NZG | 51 | 66 | 28% | 62% | 111 | 121 | 120.5 | 9% | 57% | 42 | 39 | -7% | 52% |
| GY | 55 | 65 | 17% | 57% | 107 | 116 | 115.4 | 9% | 75% | 36 | 33 | -8% | 54% |
| WL | 52 | 41 | -21% | 54% | 92 | 112 | 111.6 | 22% | 73% | 35 | 33 | -7% | 54% |
| XC | 43 | 31 | -28% | 47% | 100 | 108 | 107.3 | 8% | 52% | 33 | 29 | -12% | 55% |
| HR | 26 | 11 | -56% | 70% | 124 | 105 | 104.4 | -15% | 47% | 27 | 22 | -19% | 51% |
| CP | 42 | 28 | -34% | 58% | 96 | 91 | 90.3 | -5% | 77% | 33 | 32 | -1% | 54% |
| ATZX | 56 | 62 | 10% | 55% | 105 | 107 | 106.2 | 1% | 68% | 33 | 31 | -4% | 54% |
| GC | 56 | 42 | -25% | 58% | 106 | 107 | 106.4 | 0% | 59% | 43 | 37 | -14% | 52% |

WSXG: Wanshouxigong; DL: Dingling; DS: Dongsi; TT:Tiantan; NZG:Nongzhanguan;
GY: Guanyuan; WL: Wanliu; XC:Xincheng; HR:Huairou; CP:Changping;
ATZX:Aotizhongxin; GC:Gucheng
Sim*: $O_3$ concentration in uptake coefficient increased by a factor of 10

**Comment 3:**  Figure 3(e, f) do not show the higher OH plot in Het case, so the
statement in line 366 can't get the conclusion that high ANO3 is attribute to the elevated
OH, although other studies presented the same conclusion. By the way, the photolysis
of ClNO2 also release NO2 and enhance the reaction of OH + NO2. It would be good
if the increasing of daytime NO2 and OH by ClNO2 photolysis can be quantified and
used to assess the contribution from NO2 and OH.

**Response #3**: We have supplemented the spatial distribution of diurnal OH
concentration (panel (a)) and difference ((HET/BASE)/BASE, panel (b)) averaged from
11-15 June 2017. Which can reflect that the high ANO3 is attribute to the elevated OH.
We add the Figure R1 in SI.
As the reviewer's representation, the photolysis of $ClNO_2$ also release $NO_2$ and enhance
the reaction of OH + $NO_2$. However, it's hard to quantify its contribution to $NO_2$ in the
model. But we think this process can be ignored because the $NO_2$ generated by $ClNO_2$
is rather low (because the $ClNO_2$ is rather low, less than 1ppb). By contrast, the
atmospheric $NO_2$ concentration is reaching up to 20~40ppb.

[Figure]

Figure R1. Spatial distributions of episode-average OH concentration (a) and the
difference (b) from 11 to 15 June 2017. Unit: $10^6$ molecules $cm^{-3}$

**Comment 4:**  L89, high Cl- do not represent high ClNO2 yield, which may be subject
to the aerosol liquid water content, organics and so on, please change to "When ClNO2
yield is high…".

**Response #4**: The sentence 'When $Cl^-$ is enough, this reaction leads to lower nitrate
concentrations than reaction R5.' in line 89 is deleted.

**Comment 5:**  Line 229, Eqs. 8, change to γOH.

**Response #5**: Revised.

**Comment 6:** Line 321, Bracket is not complete.

**Response #6**: Revised

**Comment 7:** Line 328, Tham et al., 2019 change to 2018.

**Response #7**: Revised.

**Comment 8:** Line 718-719, the font size is bigger than other words.

**Response #8**: Revised.

**Comment 9:** Line 332-336, the long sentence is difficult to read. Please divide into two sentences, "… and the larger constant N2O5 leads to…" change to ". The application of large and fixed N2O5 uptake coefficient … "

**Response #9**: We have revised following the reviewer's option.

**Comment 10:** Line 377, subscript of "PM2.5".

**Response #10**: Revised.

**Comment 11:** Figure 4 should add the base case percentage like HET in the left edge.

**Response #11**: The left edge in Figure 4 presents the contribution of heterogeneous reaction and $HNO_3$ partitioning to nitrate formation, which is concentration contribution, not the percentage.

**Report #2**

**Comment 1:** There is one thing I would like to mention, although it will not change the results and the conclusion of the present study.
In Response 35, the authors stated that "HONO photolysis affects the OH level just a few hours in the morning and can be neglected", which is not correct. It has been well established that HONO is produced both in nighttime and in the daytime, and HONO is the predominant source of OH radical during daytime (not just a few hours in the morning) in the polluted environment, e.g. Fu et al., 2019 (Fig 2 and 5) and the reference therein.

Fu, X., Wang, T., Zhang, L., Li, Q., Wang, Z., Xia, M., Yun, H., Wang, W., Yu, C., Yue, D. and Zhou, Y., 2019. The significant contribution of HONO to secondary pollutants during a severe winter pollution event in southern China. Atmospheric Chemistry and Physics, 19(1), pp.1-14.

**Response #1**: Thanks for the reviewer correcting our addressing. In original CMAQ, HONO is produced by the reaction $NO_2$ with $H_2O$. However, this study improves this reaction to produce ClNO and nitrate, not involve HONO. From Fu's work, there are some other HONO sources, which may increase the OH level in daytime. But we don't intend to improve these to our study since our work focus on the impact of chlorine
heterogeneous reaction on the nitrate.

[revised manuscript text omitted]